# Sensitivity analysis of mesoscale simulations to physics parameterizations over the Belgian North Sea using WRF-ARW

Adithya Vemuri[1,2,3], Sophia Buckingham[1], Wim Munters[1], Jan Helsen[2], and Jeroen van Beeck[1]

[1]Department of Environmental and Applied Fluid Dynamics, von Karman Institute for Fluid Dynamics, Waterloosesteenweg 72, 1640 Sint-Genesius-Rode, Belgium
[2]Department of Mechanical Engineering, Vrije Universiteit Brussel, Boulevard de la Plaine 2, 1050 Ixelles, Belgium
[3]SIM vzw, Technologiepark 48, 9052 Zwijnaarde, Belgium

**Correspondence:** Adithya Vemuri (adithya.vemuri@vki.ac.be)

**Abstract.**

The Weather Research and Forecasting (WRF) model offers a multitude of physics parameterizations to study and analyze the different atmospheric processes and dynamics that are observed in the Earth's atmosphere. However, the suitability of a WRF model setup is known to be highly sensitive to the type of weather phenomena and the type and combination of physics parameterizations. A multi-event sensitivity analysis is conducted to identify general trends and suitable WRF physics setups for three extreme weather events identified to be potentially harmful for the operation and maintenance of wind farms located in the Belgian offshore concession zone. The events considered are Storm Ciara on 10 February 2020, a low-pressure system on 24 December 2020, and a trough passage on 27 June 2020. 12 WRF simulations per event are performed to study the effect of the update interval of lateral boundary conditions and different combinations of physics parameterizations (PBL, cumulus, and microphysics). Specifically, the update interval of ERA5 lateral boundary conditions is varied between hourly and 3-hourly. Physics parameterizations are varied between three PBL schemes (MYNN, scale-aware Shin-Hong, and scale-aware Zhang), four cumulus schemes (Kain-Fritsch, Grell-Dévényi, scale-aware Grell-Freitas, and multi-scale Kain-Fritsch), and three microphysics schemes (WSM5, Thompson, and Morrison). The simulated wind direction and wind speed are compared qualitatively and quantitatively (using MAE) to operational SCADA data. Overall, a definitive best-case setup common to all three events is not identified in this study. For wind direction and wind speed, the best-case setups are identified to employ scale-aware PBL schemes. These are most often driven by hourly update intervals of lateral boundary conditions as opposed to 3-hourly, although it is only in the case of storm Ciara that significant differences are observed. Scale-aware cumulus schemes are identified to produce better results when combined with scale-aware PBL schemes, specifically for Storm Ciara and the trough passage cases. However, for the low-pressure system case this trend is not observed. No clear trend in utilizing higher-order microphysics parameterization considering the combinations of WRF setups in this study is found in all cases. Overall, the combination of PBL, cumulus, and microphysics schemes is found to be highly sensitive to the type of extreme weather event. Qualitatively, precipitation fields are found to be highly sensitive to model setup and the type of weather phenomena.

# 1 Introduction

Extreme weather phenomena such as low-level jets, fast changes in wind direction, extreme wind shear (Kalverla et al., 2017; Aird et al., 2021), wind ramps (Gallego-Castillo et al., 2015), and storms (Solari, 2020) are capable of causing severe dynamic loading on wind turbines (Negro et al., 2014; AbuGazia et al., 2020; Chi et al., 2020). Furthermore, precipitation associated with these phenomena can lead to early blade degradation through leading-edge erosion (Law and Koutsos, 2020). As such, these extreme weather events (EWEs) play a significant role in wind turbine operational lifetime and must be considered at the design stage to ensure safe estimates of ultimate and fatigue loading. Such events may also cause sudden changes in power production leading to grid imbalance and economic losses. Therefore, accurate modeling and forecasting of such EWEs are crucial for the operation of onshore and offshore wind farms. Typically, numerical weather prediction (NWP) models are utilized to identify, study, and analyze such extreme weather phenomena. Recent developments in NWP models pave the way towards high resolution weather forecasts, thus enabling operational use for wind energy applications (Dudhia, 2014; Bauer et al., 2015). This study utilizes the public domain Weather Research and Forecasting - Advanced Research WRF (WRF-ARW) model developed by the National Center for Atmospheric Research (Skamarock et al., 2019; Powers et al., 2017). The WRF model represents a multitude of atmospheric processes and dynamics such as the distribution of fluxes within the planetary boundary layer (PBL), the determination of cloud ensembles and compensating subsidence for convective cumulus systems, and the evolution of hydrometeor species. Therein, an array of physics parameterizations and model parameters are available to adequately represent a local weather system. Nonetheless, WRF simulations are found to be highly sensitive to the type and combination of physics schemes, the location and the type of weather event, and the lateral boundary conditions (LBCs).

Sensitivity analyses are typically conducted to identify the optimal combination of physics schemes for a specific location (e.g., Efstathiou et al. 2013; Santos-Alamillos et al. 2013; Kala et al. 2015). This type of investigation has not been performed for the Belgian North Sea. Furthermore, to the authors' best knowledge, no previous studies have looked at potentially harmful EWEs from a wind farm perspective as experienced by the machines themselves. Therefore, this sensitivity analysis aims to address this gap in research. The analysis presented in this paper assesses the impact of a wide range of physics parameterizations for PBL, cumulus and microphysics, and length of the update interval of LBCs on the simulated wind direction and wind speed.

WRF physics parameterizations for the PBL, cumulus, and microphysics comprise a multitude of large-scale and sub-grid scale modeling techniques. For the PBL and cumulus, these are primarily divided into scale-aware and non-scale-aware parameterizations. The scale-aware parameterizations aim to better represent convective and turbulent fluxes at the so-called gray-zone resolutions, i.e., for refined horizontal grid spacings which are on the verge of allowing partial resolution of these fluxes rather than fully parameterizing them (Wyngaard, 2004; Hong and Dudhia, 2012). The following paragraphs briefly discuss the state-of-the-art physics parameterizations of the PBL, cumulus, and microphysics.

Concerning the parameterization of boundary-layer turbulence, traditional PBL schemes rely on the assumption of horizontal homogeneity to redistribute surface fluxes vertically within the atmospheric boundary layer. However, for horizontal grid spacings of around 1 km or finer, 3D atmospheric turbulence becomes partially resolved, violating this basic assumption

employed by classical 1D PBL schemes. The gray-zone modeling challenge for PBL turbulence has led to the development of scale-aware PBL schemes which partially resolve turbulent mixing at gray-zone resolutions as a function of the grid spacing. This work considers three PBL parameterizations: the non-scale-aware 1D Mellor-Yamada-Nakanishi-Niino (MYNN) scheme,

the scale-aware 1D Shin-Hong (SH) scheme, and the scale-aware 3D Zhang scheme. The MYNN PBL scheme (Nakanishi and Niino, 2006) is a 1D turbulence kinetic energy prediction scheme that solves for a vertical eddy viscosity profile in a grid column. The SH PBL scheme (Shin and Hong, 2015) is a scale-aware 1D diagnostic non-local scheme representing non-local transport by large eddies in the atmospheric boundary layer. The SH PBL scheme modifies the YSU PBL scheme (Hong et al., 2006) for sub-kilometer transition scales by reducing the strength of the non-local term with decreasing horizontal grid spacing,

assuming gradual resolution of the largest eddies. The Zhang 3D PBL scheme (Zhang et al., 2018) extends the 3D turbulent kinetic energy based closure by Deardorff (Deardorff, 1980) to gray-zone resolutions, using partitioning functions derived from a reference large eddy simulation. While the SH PBL scheme has been found to outperform conventional PBL formulations for desert convective boundary layers (Xu et al., 2018) and for the western Great Plains of the United States (Doubrawa and Muñoz-Esparza, 2020), its interaction with cumulus and microphysics options is yet to be tested for extreme weather in coastal

environments featuring strong interaction between PBL and convective cumulus processes.

The cumulus parameterizations represent the ensemble effects of convective clouds with statistical effects of moist convection and convective rainfall within a grid column. Cumulus schemes are further divided into mass-flux type and adjustment type. The mass-flux type schemes aim to minimize the convective available potential energy within a grid column by converting it into compensating subsidence, a combination of vertical advection, moisture, and temperature. The current work considers

the Kain-Fritsch (KF), the multi-scale Kain-Fritsch (msKF), the Grell-Dévényi 3D ensemble (GD-3D), and the scale-aware Grell-Freitas (GF) cumulus schemes, to evaluate the performance of WRF across the convective gray-zone. The KF cumulus scheme (Kain, 2004) is a commonly used 1D mass-flux type scheme that considers deep and shallow convection. The scheme includes hydrometeor detrainments from clouds, rain, ice, and snow. The scheme is designed to run at a horizontal grid spacing of 25 km and coarser. The msKF cumulus scheme (Zheng et al., 2016) updates the KF cumulus scheme to convective gray-zone

resolutions at horizontal grid spacings of 10 km and coarser. The GD-3D cumulus scheme (Grell and Dévényi, 2002) relies on combining ensemble and data assimilation techniques to represent the local convection and provides adjustable parameters for further calibration of the scheme. The GF cumulus scheme (Grell and Freitas, 2014) is an adjustment type parameterization that redistributes compensating subsidence derived from GD-3D to neighboring grid cells using a Gaussian distribution function and adapts the scale-aware cloud representations from Arakawa et al. (2011). The GF cumulus scheme is designed and tested

for horizontal grid spacings of 5 km and coarser. A study by Jeworrek et al. (2019) highlights the importance of choosing an appropriate cumulus parameterization to accurately represent precipitation, particularly in the convective gray-zone.

Microphysics parameterizations emulate the processes of moisture removal from the atmosphere by modeling hydrometeor distributions based on thermodynamic and kinematic fields defined in WRF. These schemes determine the spatial distribution of precipitation and the 3D distribution of hydrometeor mass and latent heat. The most commonly used microphysics schemes are

90 the so-called bulk schemes. These constitute a mathematical distribution of hydrometeor number concentration versus particle size using either a negative exponential or a gamma distribution. Bulk type microphysics parameterizations are further divided

by order of complexity and number of tunable parameters, which define the moments and intercepts used by the aforementioned distributions. The microphysics schemes considered for this sensitivity analysis are the WRF Single-Moment five-class scheme (WSM5) (Hong et al., 2004) representing five classes of hydrometeor species, the Thompson single-moment (except ice) six-class scheme (Thompson et al., 2008), and the Morrison double-moment six-class scheme (Morrison et al., 2009). In this respect, Hong and Lim (2006) illustrated the advantages in including a greater number of hydrometeor species in microphysical representations to better predict precipitation fields. Similar results were observed in a recent study by Jeworrek et al. (2019), calling for microphysics parameterizations with greater fidelity in hydrometeor representation. The study finds that higher-order microphysics schemes, such as Morrison and Thompson, in combination with scale-aware cumulus parameterizations, such as multi-scale KF and GF schemes, accurately reproduce precipitation.

In the context of offshore wind energy applications, various sensitivity studies have been conducted with the aim of determining a universal best-case WRF setup for assessing the local weather systems (Hahmann et al., 2015; Giannakopoulou and Nhili, 2014; Carvalho et al., 2012). Therein, the literature presents equivocal findings from a multitude of sensitivity analyses conducted at various locations around the planet, illustrating a strong dependence of WRF simulations to the type and combination of physics parameterizations, the initial and LBCs, the horizontal and vertical grid spacing, and the location and type of weather phenomenon. For instance, comparing wind power production to observational data, Hahmann et al. (2015) study the long-term sensitivity of simulated WRF offshore climatology evaluated against wind LiDAR observations, indicating a strong sensitivity to PBL parameterizations and the spin-up period, and an insensitivity to global reanalysis and vertical grid spacing considered in the WRF model. Similarly, Carvalho et al. (2014) indicate a close dependency on PBL and surface layer parameterizations studying different physics combinations, that may lead to increased accuracy depending on the prognostic variables of interest. Cunden et al. (2018) performed a sensitivity analysis considering different combinations of non-scale-aware PBL, cumulus, and microphysics parameterizations (despite kilometer-range grid spacing) for the island of Mauritius under clear and extreme weather. The study was able to identify a best-case WRF setup suitable for accurately simulating both cases. In contrast, the study by Islam et al. (2015) for the Haiyan tropical cyclone over the west Pacific Ocean did not identity a suitable combination of WRF physics to best reproduce the extreme weather event. Similarly, for the European continent, studies by García-Díez et al. (2013), Stergiou et al. (2017), and Mooney et al. (2013) have conducted long-term sensitivity analyses indicating a wide array of possible combinations of physics parameterizations depending on the type of weather phenomenon, the season, and the time period to simulate within the diurnal cycle.

The optimal selection of WRF physics parameterizations remains an important and open challenge to accurately simulate weather phenomena. The current study quantifies the sensitivity of WRF simulation results to physics parameterizations and model setup to identify best suitable combinations for modeling three EWEs detected from SCADA data collected at the Belgian offshore wind farms. This multi-variant multi-event sensitivity analysis considers 12 physics combinations comprising three PBL schemes, four cumulus schemes, three microphysics schemes, and hourly versus 3-hourly update intervals of LBCs. The remainder of this article is structured as follows. Firstly, a description of EWEs is introduced in Sect. 2. Next, the numerical methodology and modeling setup are introduced in Sect. 3. The simulation results and discussions are presented in Sect. 4. Lastly, conclusions and future prospects are presented in Sect. 5.

## 2 Description of the events

The selection of the events in this study is motivated by the occurrence of fast changes in wind direction accompanied by severe yaw misalignment leading to significant power loss as observed by a Belgian offshore wind farm in the North Sea. The methodology utilized to identify these events modifies the approach defined by Hannesdóttir and Kelly (2019) to include yaw misalignment. The wavelet analysis considers a minimum threshold to identify anomalous changes in wind direction accompanied by severe yaw misalignment experienced by several wind turbines. Severe yaw misalignment potentially has adverse effects on the operational lifetime and fatigue loading of a wind turbine (Wan et al., 2015; Bakhshi and Sandborn, 2016; Laino and Hansen, 1998; Damiani et al., 2018), highlighting its importance and relevance in this study. The SCADA analysis for the identification of these events includes confidential error codes and data that are protected under a non-disclosure agreement, therefore no further details can be provided herein.

Three case studies are considered in this sensitivity analysis, namely, Storm Ciara on 10 February 2020, a low-pressure system on 24 December 2020, and a trough passage on 27 June 2020. The radar data presented herein is not publicly available, but was retrieved through a bilateral agreement with the Royal Meteorological Institute of Belgium (RMI-B). A brief synopsis of these events is presented in the following sub-sections.

### 2.1 Case study 1: Storm Ciara

Storm Ciara was one of the first extratropical cyclones to hit the European continent in the year 2020, occurring on 10 February 2020 over the Belgian North Sea. Storm Ciara originated in the Atlantic Ocean, moving from the North American continent (starting 3 February 2020) to the European continent (16 February 2020). Storm Ciara swept across the majority of western Europe including the United Kingdom and Norway, bringing in heavy precipitation and strong winds with a maximum recorded wind gust of 219 km h$^{-1}$ at Cap Corse, Corsica, France[1]. Over Belgium, the RMI-B[2] reported wind gusts of up to 115 km h$^{-1}$ in Ostend, located at the Belgian coast, with heavy precipitation accompanied by strong winds and thunderstorms.

During the early hours of Storm Ciara on 10 February 2020, an offshore wind farm recorded fast changes in wind direction accompanied by severe yaw misalignment and concentrated rainfall. An RMI-B radar snapshot at 04:40 UTC is presented in Fig. 1a, illustrating the presence of a bow-echo moving from the British Isles to Belgium, an indication of a possible micro-burst phenomenon (Fujita, 1978). Synoptic maps by the Royal Netherlands Meteorological Institute (RNMI[3]) presented in Fig. 2a indicate a trough passage during this period. Further, precipitation data from a wind profiler located within the wind farm highlights fast changes in wind direction accompanied by sudden precipitation during the period of interest at 04:40 UTC, presented in Fig. 3.

[1]https://www.meteo-paris.com/actualites/retro-meteo-2020-les-evenements-climatiques-marquants-en-france, website consulted on 21 April 2022.
[2]https://www.meteo.be/nl/info/nieuwsoverzicht/storm-ciara, website consulted on 21 April 2022.
[3]https://www.knmi.nl/nederland-nu/klimatologie/daggegevens/weerkaarten, website consulted on 21 April 2022.

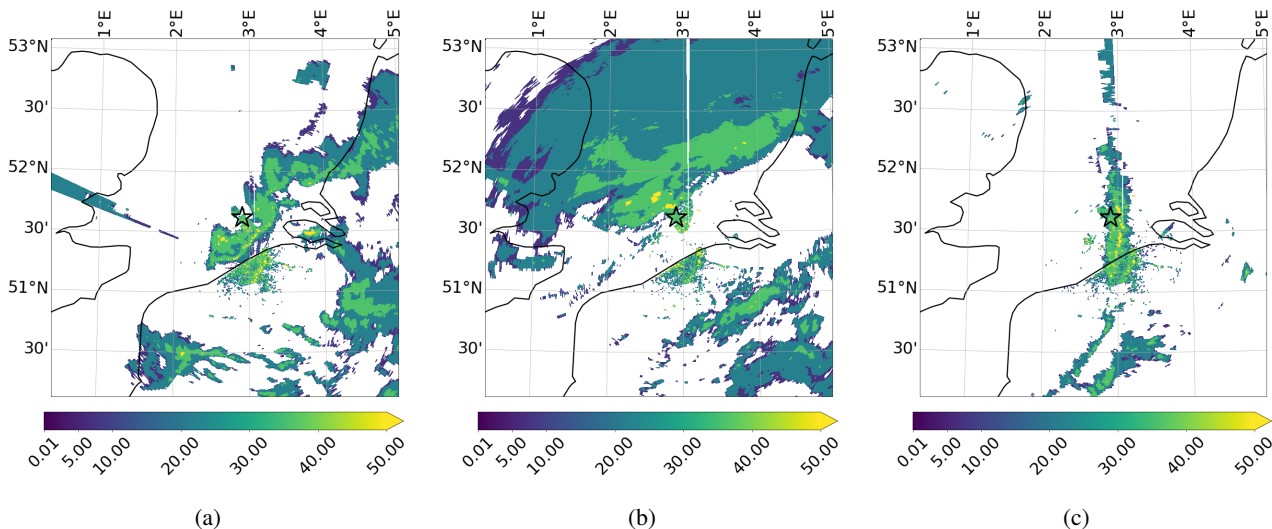

**Figure 1.** Observed precipitation rate in mm h$^{-1}$ provided by a C-band Doppler radar located in Jabbeke on the Belgian coast. The star in the plots represents the offshore wind farm of interest. For the meteorological events: (a) Storm Ciara on 10 February 2020 at 04:40 UTC. (b) Low-pressure system on 24 December 2020 at 02:00 UTC. (c) Trough passage on 27 June 2020 at 15:30 UTC.

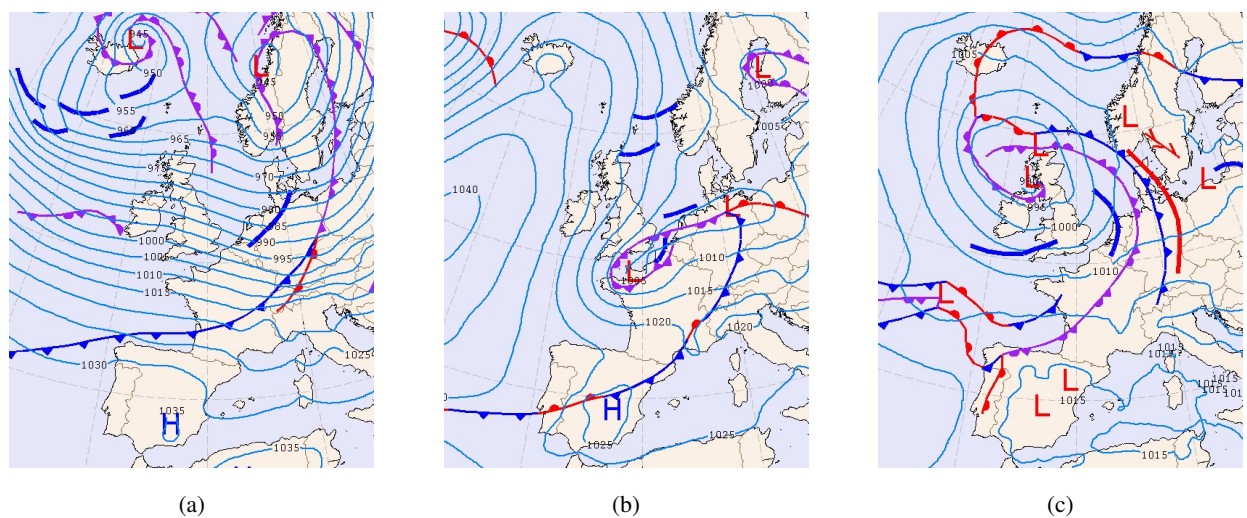

**Figure 2.** Synoptic maps provided by RNMI. (a) Storm Ciara on 10 February 2020 at 06:00 UTC. (b) Low-pressure system on 24 December 2020 at 00:00 UTC. (c) Trough passage on 27 June 2020 at 18:00 UTC.

## 2.2   Case study 2: Low-pressure system

On 24 December 2020, the Belgian offshore wind farms observed heavy precipitation accompanied by fast changes in wind direction. Synoptic maps presented in Fig. 2b indicate the presence of a low-pressure system over the English Channel and

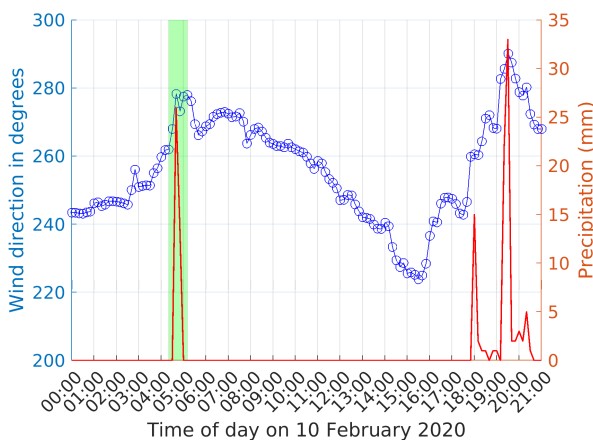

**Figure 3.** Precipitation observed by the offshore wind farm plotted against 10-min averaged wind direction (SCADA data). The highlighted the period of interest, in green, at 04:40 UTC is observed to accompany sudden precipitation.

Normandy. Radar observations from the RMI-B indicate large precipitation cells over the Belgian North Sea, presented in Fig. 1b. SCADA data records fast changes in wind direction of $100°$ at 02:00 UTC accompanied by severe yaw misalignment and precipitation.

### 2.3 Case study 3: Trough passage

On 27 June 2020, the Belgian offshore wind farms experienced fast changes in wind direction and sudden precipitation during the afternoon hours around 15:30 UTC. The synoptic maps provided by the RNMI indicate the presence of a low-pressure system over the British Isles, with a trough passage across the Belgian North Sea, presented in Fig. 2c. Radar observation provided by the RMI-B indicate the presence of precipitation cells over the offshore wind farms, presented in Fig. 1c. The operational SCADA data records fast changes in wind direction of $60°$ during this hour accompanied by severe yaw misalignment and precipitation.

## 3 Model setup, methodology and performance metrics

This sensitivity study considers the WRF model version 4.2.2 to simulate the case studies described in Sect. 2. The following sections describe the part of the WRF model setup common to all simulations, individual run setups used in the sensitivity study, and performance evaluation metrics for comparison to observational data. This evaluation uses operational wind farm SCADA data for its quantitative analysis of wind direction and wind speed. Additionally, radar data from RMI-B allows for a qualitative perspective on precipitation. By combining these observational datasets, the premise of this study provides a unique opportunity to investigate EWEs as experienced by an offshore wind farm to determine suitable WRF setups in the specific context of wind energy applications.

## 3.1 Common model setup

The common model parameters considered for all WRF simulations are summarized in Table 1. The baseline horizontal grid spacing of the parent domain d01 is 27 km, while the one-way nested domains are sequentially refined by a factor of three, resulting in horizontal grid spacings of 9, 3, and 1 km for d02, d03, and d04 respectively. In the vertical direction, 57 terrain-following model levels are considered with a model top pressure at 1000 Pa. The vertical velocity damping option based on the Courant–Friedrichs–Lewy condition as implemented in WRF is also turned on. A time step of 20 s is considered for parent domain d01, while the time step of nested domains is sequentially refined by a factor of three. The initial and LBCs are derived from ERA5 reanalysis (Hersbach et al., 2020). WRF simulations were initialized with a spin-up period of 24 hours for all case studies. In addition, an evaluation period of 21 hours from 00:00 to 21:00 UTC on 10 February 2020 is considered for Storm Ciara. For the low-pressure system and trough passage, an evaluation period of 6 h from 00:00 to 06:00 UTC on 24 December 2020 and 6 h from 12:00 to 18:00 UTC on 27 June 2020 are considered, respectively. The simulations have been performed as a continuous run including spin-up and evaluation periods. Therein, the selected evaluation periods adequately capture the time periods of interest for respective case studies as described in Sect. 2. The one-way nested domain configuration common to all simulations in this study is presented in Fig. 4.

The Rapid Radiative Transfer Model (RRTMG) (Iacono et al., 2008) for longwave and shortwave radiation physics is used by all simulations. Similarly, the land–surface interactions are defined by the unified Noah land surface model (Tewari et al., 2004). The PBL, cumulus, and microphysics schemes are varied amongst the mentioned options as described in Table 1.

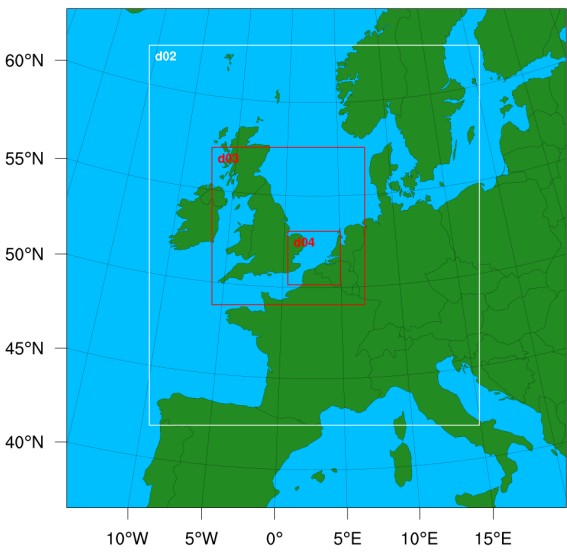

**Figure 4.** WRF nested domain configuration (one-way nesting) considered common to all simulation runs in this study.

**Table 1.** WRF model setup and common parameters for all simulation runs. The varied model settings and physics parameterizations are highlighted in italics. Scale-aware physics parameterizations are underlined.

| **Numerical setup** | |
| --- | --- |
| Nested domains (one-way nesting) | 4 |
| Horizontal grid spacing | 27 km (d01), 9 km (d02), 3 km (d03), 1 km (d04) |
| Terrain following vertical levels | 57 |
| Model top pressure | 1000 Pa |
| Time-steps for domain configuration | 20 s (d01), 6.67 s (d02), 2.22 s (d03), 0.74 s (d04) |
| Spin-up period | 24 h |
| Lateral & boundary conditions | ERA5 reanalysis |
| Evaluation time, additional to spin up | 21 h (Storm Ciara) and 6 h (Low-pressure system and trough passage) |
| *Boundary update interval* | *1h / 3h* |
| **Physics parametrizations** | |
| Radiation | RRTMG radiative |
| Land surface | unified Noah land-surface |
| *PBL* | *MYNN / Shin-Hong / Zhang* |
| *Microphysics* | *WSM5 / Thompson / Morrison* |
| *Cumulus* | *KF / GD-3D / msKF / GF* |

## 3.2 Individual run setups

In order to sufficiently categorize and distinguish the key features of different WRF physics parameterizations and options available, a combination of different simulation pairs as described in Table 2 is considered. A total of 12 WRF simulations are categorized into different simulation pairs (A – J) assigned to variations of the update interval of the LBCs, and the PBL, cumulus, and microphysics schemes. For each of the varied parameters, at least two different simulation pairs are considered. For example, simulation pairs A and B are assigned to the variation in update interval of the LBCs. More specifically, the simulation pairs considered are as follows. The sensitivity to hourly and 3-hourly update intervals of LBCs are assessed in simulation pairs A and B. Further, the sensitivity to scale-aware (SH and Zhang) and non-scale-aware (MYNN) PBL schemes is evaluated in pairs C, D, and E. The sensitivity to scale-aware (msKF and GF) and non-scale-aware (KF and GD-3D) cumulus parameterizations is evaluated in pairs F, G, and H. Given the convection-permitting resolutions of 3 km and 1 km for d03 and d04 respectively, the non-scale-aware KF model is explicitly turned off in these domains in simulation runs 2, 3, 5, and 6[4]. For the scale-aware cumulus models, this explicit deactivation is omitted, as they were specifically designed for operation on the verge of convection-permitting resolutions (Grell and Freitas, 2014; Zheng et al., 2016; Huang et al., 2020). The impact of microphysics schemes WSM5, Thompson, and Morrison is illustrated through pairs I and J. Each simulation pair justifies to serve as independent sets of simulations to judge the influence of a varied WRF parameter.

---

[4]It was verified that this approach resulted in better reproduction of precipitation cells and lower error metrics

**Table 2.** WRF simulation runs and respective simulation pairs considered for each of the varied parameter in this sensitivity analysis.

| Simulation run# | ERA5 LBCs updates | PBL scheme | Cumulus scheme | Microphysics scheme | Update interval pairs | PBL pairs | Cumulus pairs | Microphysics pairs |
|---|---|---|---|---|---|---|---|---|
| 1 | 3 h | MYNN | msKF | WSM5 | A | | | |
| 2 | 3 h | SH | KF | WSM5 | B | | | |
| 3 | 1 h | MYNN | KF | Thompson | | C | | |
| 4 | 1 h | MYNN | msKF | WSM5 | A | D | | |
| 5 | 1 h | SH | KF | WSM5 | B | | F | I |
| 6 | 1 h | SH | KF | Thompson | | C | G | |
| 7 | 1 h | SH | msKF | WSM5 | | D | F | J |
| 8 | 1 h | SH | msKF | Thompson | | | G | |
| 9 | 1 h | SH | msKF | Morrison | | | H | |
| 10 | 1h | SH | GD-3D | Morrison | | | | |
| 11 | 1 h | SH | GF | Morrison | | E | | |
| 12 | 1 h | Zhang | GF | Morrison | | | | |
| 13 | Ensemble average | | | | | | | |

## 3.3 Performance metrics and observations

The simulated wind direction and horizontal wind speed are evaluated against 10 minute averaged SCADA data from the southwestern front row of an offshore wind farm located in the Belgian North Sea. Model accuracy is assessed using a standard mean absolute error (MAE) for wind direction and wind speed. To recover a single performance metric, a so-called normalized Euclidean distance (NED) is defined by $\text{NED} = \sqrt{\text{MAE}_{WDn}^2 + \text{MAE}_{WSn}^2}$, with $\text{MAE}_{WDn}$ the normalized MAE of wind direction, and $\text{MAE}_{WSn}$ the normalized MAE of horizontal wind speed. Normalization is performed with the mean over all simulations.

Precipitation fields are qualitatively compared between WRF simulations. The simulated radar reflectivity is converted to precipitation rate using the Marshall and Palmer relation (Marshall and Palmer, 1948).

This study also evaluates the performance of an ensemble average compared to single deterministic simulation runs. The ensemble average is defined as the mean of all simulation runs considered for a given case study. In this study, ensemble members are initialized with identical initial conditions from ERA5 reanalysis. Subsequently, variability in the ensemble average is only caused by the variation in update interval of LBCs and physics parameterizations. Therein, the current definition of ensemble average differs from traditional ensemble forecasts, where variations in initial conditions are also considered, e.g., Wilks (2019).

## 4   Results and Discussion

The following Sect. 4.1, 4.2, and 4.3 present results and discussions for the overall trend in simulation results evaluated against SCADA data for Storm Ciara, the low-pressure system, and the trough passage cases, respectively. The MAE and NED are presented in performance evaluation tables under each case study. The table sequence is organized in the order of increasing complexity of the combination of physics parameterizations and shorter update interval, starting with the longer update interval of LBCs and non-scale-aware physics parameterizations, to scale-aware physics parameterizations with the shorter update intervals of LBCs. Cells are colored based on a set of 5 categories between red and green. Categories are defined to cover 20% of the range between smallest and largest values for the considered metric. In this way, results are categorized into best (green, with errors in the 20% lowest range), good (light green), average (yellow), poor (light red), and worst (dark red). In addition to this high-level assessment of setups, Sect. 4.4, 4.5, and 4.6 discuss simulation pairs addressing the influence of specific combinations of physics parameterizations and update interval of LBCs. The simulated wind direction and wind speed are quantitatively evaluated against SCADA data. Finally, Sect. 4.7 provides a synthesis of the observations in this study.

### 4.1   Case: Storm Ciara

The MAE of wind direction, horizontal wind speed, and the NED for the Storm Ciara runs is presented in Table 3. Overall, for simulation runs 2 through 12, relatively lower MAE values for wind direction and wind speed are observed, with a maximum of 9.26° and 2.72 m s$^{-1}$ respectively. Using NED as the evaluation metric, the best-case setup is determined to be simulation run 7, with a NED value very similar to the ensemble average (< 1 % difference). Run 7 uses the scale-aware SH PBL scheme coupled with the scale-aware msKF cumulus scheme, single moment five-class WSM5 microphysics scheme, and hourly ERA5 lateral boundary condition updates. In a general sense, simulation runs 7 through 10 observe the lowest overall NED. These simulation runs consider scale-aware PBL and cumulus parameterizations coupled with hourly update intervals of LBCs. A qualitative analysis of wind direction and wind speed timeseries for all simulation runs highlighting the ensemble average and the best-case setup is presented in Fig. 5. Compared to the SCADA reference data, the changes in wind direction are captured reasonably well by all runs, with the ensemble average capturing the general transience of wind direction better than the best-case setup. However, accurately capturing the variability on wind speed is found to be more challenging, as shown by the large spread among different modeling setups in the afternoon and evening hours.

### 4.2   Case: Low-pressure system

The summary of the performance evaluation metrics for the low-pressure system is presented in Table 4. The best-case setup is found to be simulation run 2, comprising scale-aware SH PBL coupled with non-scale-aware KF cumulus, WSM5 microphysics, and hourly ERA5 reanalysis data as LBCs. Unlike the case of Storm Ciara, no trend in better results for simulation runs combining scale-aware PBL, scale-aware cumulus, and higher-order microphysics is found (i.e., simulation runs 6 through 10 in Table 4). The overall trend in wind speed MAE results shows simulations using the MYNN PBL scheme, i.e., runs 1, 3, and 4, to perform poorly. Similar to the case of Storm Ciara, the best-case results are found to be very similar to the ensemble

**Table 3.** Performance metrics MAE and NED for wind direction and wind speed from all WRF simulations evaluated against SCADA data for the case of Storm Ciara. The best-case setup considering NED is simulation run 7. The minimum metric specific values are underlined.

| Simulation run# | ERA LBCs updates | PBL scheme | Cumulus scheme | Microphysics scheme | Wind direction MAE (degrees) | Wind speed MAE (m s$^{-1}$) | NED (-) |
|---|---|---|---|---|---|---|---|
| 1 | 3 h | MYNN | msKF | WSM5 | 10.46 | 3.88 | 2.08 |
| 2 | 3 h | SH | KF | WSM5 | 8.48 | 2.57 | 1.51 |
| 3 | 1 h | MYNN | KF | Thompson | 9.26 | 2.72 | 1.63 |
| 4 | 1 h | MYNN | msKF | WSM5 | 8.61 | 2.54 | 1.51 |
| 5 | 1 h | SH | KF | WSM5 | 7.68 | 2.47 | 1.41 |
| 6 | 1 h | SH | KF | Thompson | 8.37 | 2.51 | 1.48 |
| 7 | 1 h | SH | msKF | WSM5 | 6.59 | 1.78 | 1.11 |
| 8 | 1 h | SH | msKF | Thompson | 6.69 | 1.89 | 1.15 |
| 9 | 1 h | SH | msKF | Morrison | 7.17 | 1.89 | 1.20 |
| 10 | 1 h | SH | GD-3D | Morrison | 5.59 | 2.25 | 1.17 |
| 11 | 1 h | SH | GF | Morrison | 7.17 | 2.67 | 1.43 |
| 12 | 1 h | Zhang | GF | Morrison | 8.69 | 1.84 | 1.34 |
| 13 | | Ensemble average | | | 5.88 | 2.04 | 1.12 |

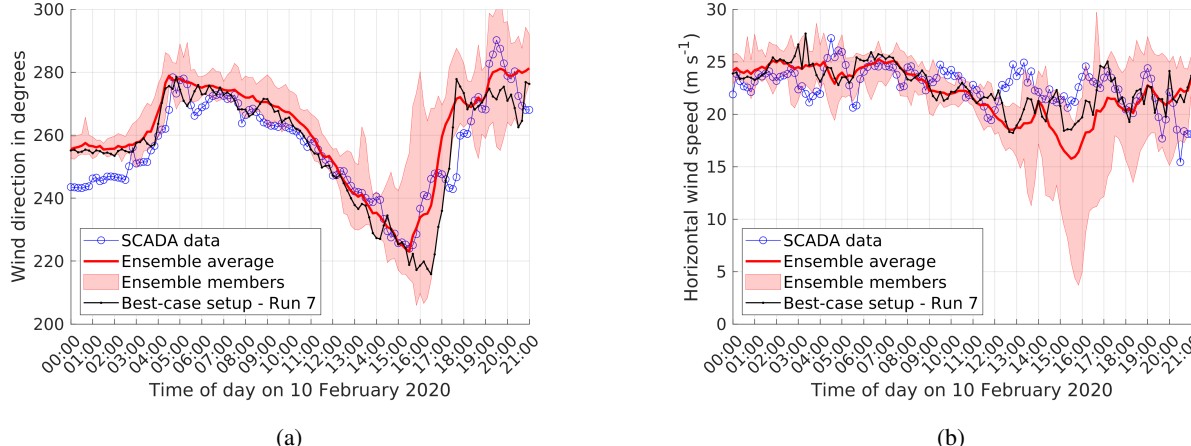

(a)                     (b)

**Figure 5.** Timeseries plots wind direction and wind speed plotted along with the ensemble average and best-case setup simulation run 7 for the case of Storm Ciara. The minimum and maximum envelope of ensemble members is highlighted in light red. (a) Wind direction. (b) Wind speed.

average, with a relative difference in NED of 3.2 %. However, it must be noted that the ensemble average tends to damp out the fast changes in wind direction, as plotted in Fig. 6 along with the best-case setup. A qualitative analysis on the timeseries indicates that all simulation runs capture the fast change in wind direction for the period of interest. However, these often exhibit a time lag compared to SCADA data.

**Table 4.** Performance metrics MAE and NED for wind direction and wind speed from all WRF simulations evaluated against SCADA data for the low-pressure system case. The best-case setup considering NED is simulation run 2. The minimum metric specific values are underlined.

| Simulation run# | ERA LBCs updates | PBL scheme | Cumulus scheme | Microphysics scheme | Wind direction MAE (degrees) | Wind speed MAE (m s$^{-1}$) | NED (-) |
|---|---|---|---|---|---|---|---|
| 1 | 3 h | MYNN | msKF | WSM5 | 12.58 | 3.96 | 1.66 |
| 2 | 3 h | SH | KF | WSM5 | 10.43 | 1.77 | 0.94 |
| 3 | 1 h | MYNN | KF | Thompson | 13.17 | 4.28 | 1.78 |
| 4 | 1 h | MYNN | msKF | WSM5 | 12.47 | 4.40 | 1.79 |
| 5 | 1 h | SH | KF | WSM5 | 13.60 | 2.19 | 1.20 |
| 6 | 1 h | SH | KF | Thompson | 21.92 | 1.95 | 1.62 |
| 7 | 1 h | SH | msKF | WSM5 | 16.92 | 2.16 | 1.37 |
| 8 | 1 h | SH | msKF | Thompson | 15.74 | 2.30 | 1.34 |
| 9 | 1 h | SH | msKF | Morrison | 19.81 | 3.12 | 1.73 |
| 10 | 1 h | SH | GD-3D | Morrison | 15.97 | 2.32 | 1.35 |
| 11 | 1 h | SH | GF | Morrison | 12.11 | 2.64 | 1.25 |
| 12 | 1 h | Zhang | GF | Morrison | 15.54 | 2.15 | 1.29 |
| 13 | Ensemble average | | | | 10.65 | 1.85 | 0.97 |

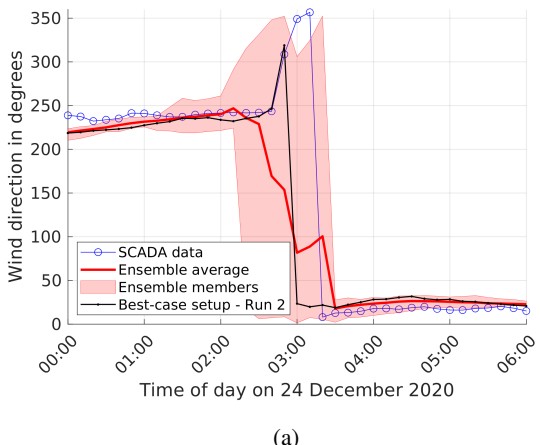

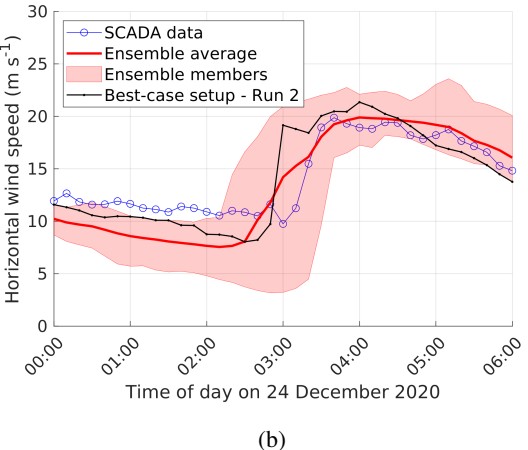

(a)                                (b)

**Figure 6.** Timeseries plots wind direction and wind speed plotted along with the ensemble average and best-case setup simulation run 2 for the low-pressure system case. The minimum and maximum envelope of ensemble members is highlighted in light red. (a) Wind direction. (b) Wind speed.

## 4.3 Case: Trough passage

The performance evaluation metrics for the trough passage are presented in Table 5. Overall, the considered WRF setups are found to be highly sensitive to the combinations and type of physics parameterizations. Overall, MAE values are found to be higher than for the other two cases, indicating that accurately predicting wind direction and wind speed is more challenging

for this particular event. No clear trend in any combinations of the sequence of simulation runs is found, in contrast to Storm Ciara and the low-pressure system. Interestingly, the best-case setup for the low-pressure system case performed the worst for the trough passage case. Considering NED as the evaluation metric, the best-case setup is observed to be run 12 by a significant margin. Run 12 uses the scale-aware Zhang 3D PBL scheme, the scale-aware GF cumulus scheme coupled with Morrison microphysics and hourly ERA5 LBCs updates. Simulation timeseries including the ensemble average and best-case setup are presented in Fig. 7. Qualitatively, simulation runs 1 through 11 underpredict the fast changes in wind direction for the evaluation period, whereas wind speeds are underpredicted by all runs. Due to the joint poor performance and persistent offsets compared to SCADA data for all simulations except run 12, the ensemble average does not yield any better match to the data.

**Table 5.** Performance metrics MAE and NED for wind direction and wind speed from all WRF simulations evaluated against SCADA data for the case of the trough passage. The best-case setup considering NED is simulation run 12. The minimum metric specific values are underlined.

| Simulation run# | ERA LBCs updates | PBL scheme | Cumulus scheme | Microphysics scheme | Wind direction MAE (degrees) | Wind speed MAE (m s$^{-1}$) | NED (-) |
|---|---|---|---|---|---|---|---|
| 1 | 3 h | MYNN | msKF | WSM5 | 15.12 | 4.22 | 1.39 |
| 2 | 3 h | SH | KF | WSM5 | 19.27 | 5.27 | 1.75 |
| 3 | 1 h | MYNN | KF | Thompson | 16.04 | 3.47 | 1.32 |
| 4 | 1 h | MYNN | msKF | WSM5 | 11.64 | 4.46 | 1.28 |
| 5 | 1 h | SH | KF | WSM5 | 17.95 | 4.51 | 1.57 |
| 6 | 1 h | SH | KF | Thompson | 19.84 | 4.60 | 1.68 |
| 7 | 1 h | SH | msKF | WSM5 | 15.05 | 5.31 | 1.57 |
| 8 | 1 h | SH | msKF | Thompson | 16.34 | 5.02 | 1.58 |
| 9 | 1 h | SH | msKF | Morrison | 10.97 | 3.78 | 1.13 |
| 10 | 1 h | SH | GD-3D | Morrison | 17.73 | 4.57 | 1.57 |
| 11 | 1 h | SH | GF | Morrison | 15.03 | 3.87 | 1.33 |
| 12 | 1 h | Zhang | GF | Morrison | 7.43 | 3.11 | 0.87 |
| 13 | | Ensemble average | | | 14.77 | 4.34 | 1.39 |

## 4.4 Update interval of lateral boundary conditions: Simulation pairs A and B

The effect of varying the update interval of ERA5 LBCs between hourly and 3-hourly is investigated in this section. Simulation pairs A and B represent four WRF setups for each case study. Figure 8 shows the results for simulation pair A. Error bars indicate one standard error of the sample mean. Starting with wind direction (Fig. 8a), hourly update intervals of ERA5 LBCs are observed to perform better for Storm Ciara and the trough passage. For the low-pressure system, lower MAE is observed for hourly reanalysis data however, these values lie well within the standard error bars, therefore leading to inconclusive overall results. For wind speed (Fig. 8b), a significant improvement on MAE is observed when hourly reanalysis data is used for Storm Ciara, with a 34 % reduction compared to 3-hourly data. In contrast, for the low-pressure system and the trough passage, 3-hourly reanalysis data produces lower MAE, however these values lie within the standard error. Overall, for simulation pair A,

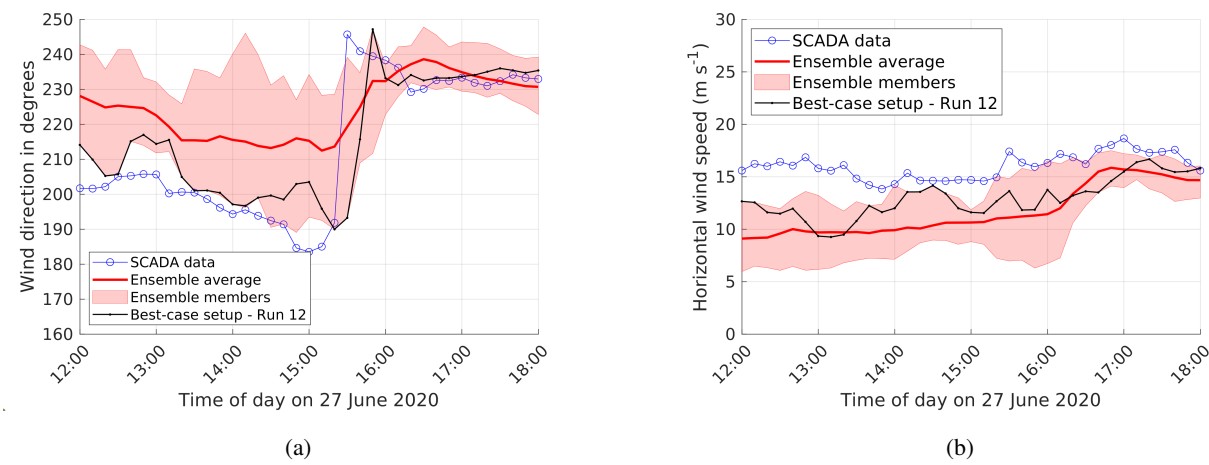

(a)                                                                          (b)

**Figure 7.** Timeseries plots wind direction and wind speed plotted along with the ensemble average and best-case setup simulation run 12 for the case of trough passage. The minimum and maximum envelope of ensemble members is highlighted in light red. (a) Wind direction. (b) Wind speed.

a distinction in better performance for hourly reanalysis is observed for Storm Ciara, however no significant benefit is observed for the other two cases.

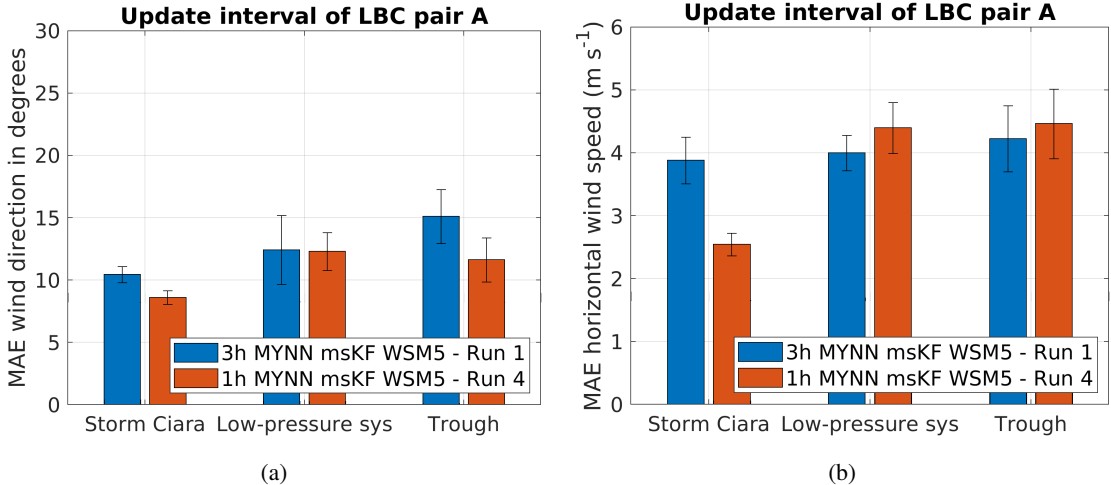

(a)                                                                          (b)

**Figure 8.** Performance evaluation for simulation pair A considering change in update interval of LBCs, as described in Table 2. Error bars indicate one standard error of the sample mean. (a) MAE comparison for wind direction. (b) MAE comparison for wind speed.

Similarly, Fig. 9 shows the results for simulation pair B. The MAE comparison for wind direction is presented in Fig. 9a, indicating statistically inconclusive results for all cases. Similarly, MAE results for wind speeds are presented in Fig. 9b,

indicating inconclusive results for Storm Ciara and the low-pressure system, yet a better performance with hourly reanalysis data in the case of the trough passage.

To summarize the overall inferences from both simulation pairs, hourly updates of LBCs do not systematically lead to higher accuracy, although improvements are observed for certain combination of events and wind variables. Therefore, more frequent updates of LBCs may prove advantageous when trying to capture certain fast transient weather events.

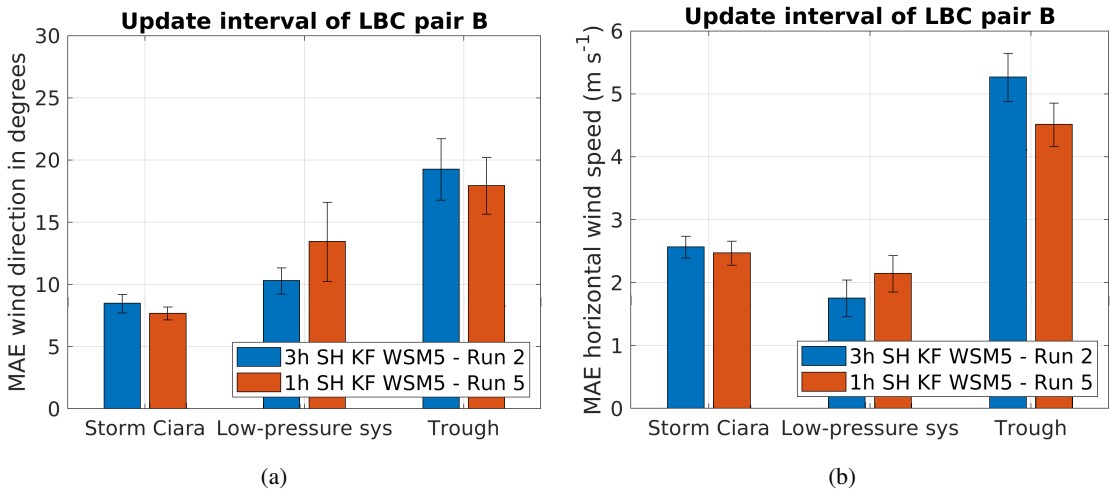

(a)          (b)

**Figure 9.** Performance evaluation for simulation pair B considering change in update interval of LBCs, as described in Table 2. Error bars indicate one standard error of the sample mean. (a) MAE comparison for wind direction. (b) MAE comparison for wind speed.

## 4.5 Planetary boundary layer: Simulation pairs C, D, and E

In this section, the influence of using classical non-scale-aware PBL schemes versus scale-aware PBL schemes is elaborated in simulation pairs C, D, and E. More specifically, the standard MYNN scheme is compared to the scale-aware 1D SH and scale-aware 3D Zhang PBL schemes. Simulation pairs C and D compare the influence of MYNN and SH PBL schemes and simulation pair E compares SH and Zhang PBL schemes.

Figure 10 presents the consolidated results for simulation pair C. First, considering wind direction MAE (Fig. 10a), the SH PBL scheme performs better for the case of Storm Ciara. In contrast, for the low-pressure system and the trough passage, the MYNN PBL scheme shows better performance. Considering wind speed (Fig. 10b), no conclusive set of inferences are drawn for the case of Storm Ciara, as the lower MAE by SH lies within the range of the standard error. For the low-pressure system, the SH PBL scheme outperforms the MYNN scheme in terms of wind speed, reversing the trend found for wind direction. For

the trough passage, MYNN wind speeds outperform SH. Overall, for the simulation pair C, no clear conclusions can be drawn for Storm Ciara and the low-pressure system cases. In contrast, better performance by the MYNN PBL scheme is observed for the trough passage.

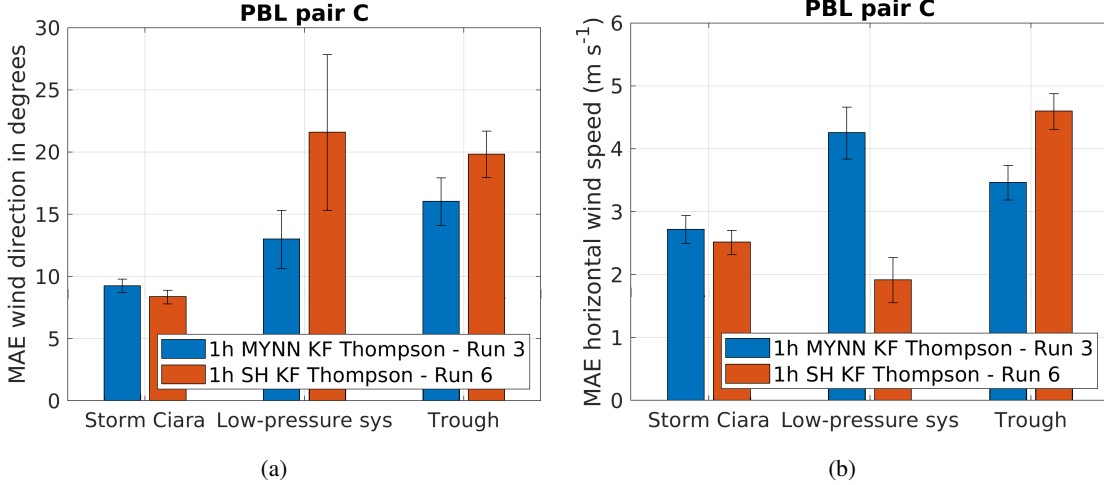

(a)                                                    (b)

**Figure 10.** Performance evaluation for simulation pair C considering a change in PBL scheme, as described in Table 2. Error bars indicate one standard error of the sample mean. (a) MAE comparison for wind direction. (b) MAE comparison for wind speed.

Figure 11 illustrates MAEs for simulation pair D. Starting with wind direction (Fig. 11a), a clear advantage in utilizing SH PBL is observed for the case of Storm Ciara. However, this distinction is not observed for the low-pressure system. The MYNN PBL scheme performs better than the SH PBL scheme for the trough passage. This trend is found similar to simulation pair C. Considering wind speeds (Fig. 11b), SH outperforms MYNN by a significant margin for the case of Storm Ciara. A similar distinction is observed for the low-pressure system. However, similar to pair C, for the trough passage this trend is reversed.

Overall, for the simulation pairs C and D, a distinctly better performance by the SH PBL scheme is observed for the case of Storm Ciara. However, little to no conclusions can be drawn for the low-pressure system. For the trough passage, better performance is observed for MYNN PBL scheme.

To further distinguish between scale-aware PBL schemes of different complexity, Figure 12 presents results for simulation pair E, comparing the SH and the Zhang PBL schemes. The wind direction MAE results (Fig. 12a) show an advantage in using SH for the case of Storm Ciara. However, this distinction is not statistically significant for the low-pressure system. The Zhang PBL scheme outperforms the SH PBL scheme by a significant margin for the trough passage. Considering wind speed (Fig. 12b), the Zhang scheme outperforms the SH scheme for Storm Ciara and the trough passage. However, this trend is not statistically significant for the low-pressure system case. The trough passage observes better performance by the Zhang PBL scheme. Overall, for simulation pair E, no clear distinction in better performance for both wind direction and wind speed combined is observed for SH or Zhang PBL schemes for Storm Ciara and the low-pressure system cases. However, for the case of the trough passage, a clear advantage in using the Zhang PBL scheme is observed.

A qualitative comparison of timeseries for the low-pressure system and trough passage cases is presented in Fig. 13b, indicating better performance by the Zhang PBL scheme to capture the transience in wind direction. However, this qualitative advantage is not observed in the case of Storm Ciara (not further plotted here).

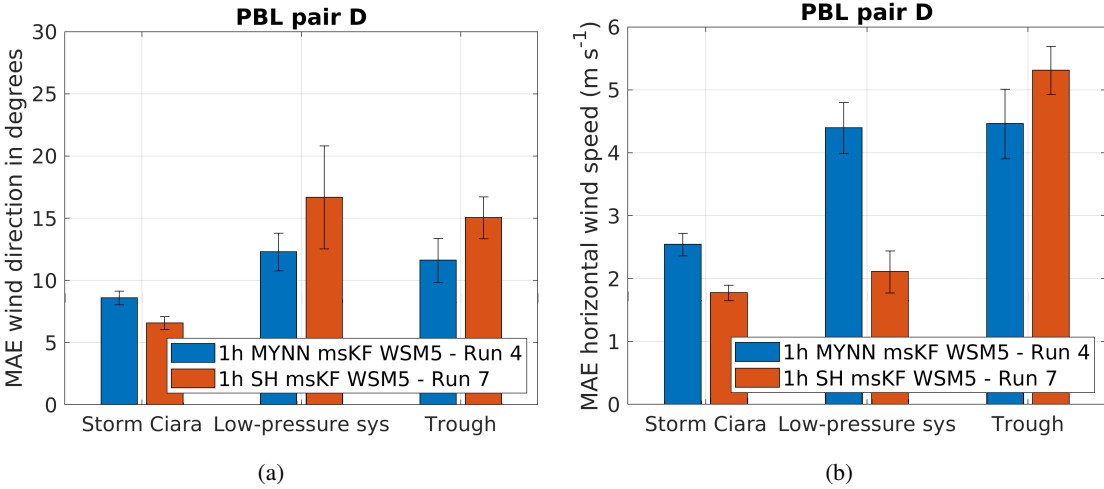

**Figure 11.** Performance evaluation for simulation pair D considering a change in PBL scheme, as described in Table 2. Error bars indicate one standard error of the sample mean. (a) MAE comparison for wind direction. (b) MAE comparison for wind speed.

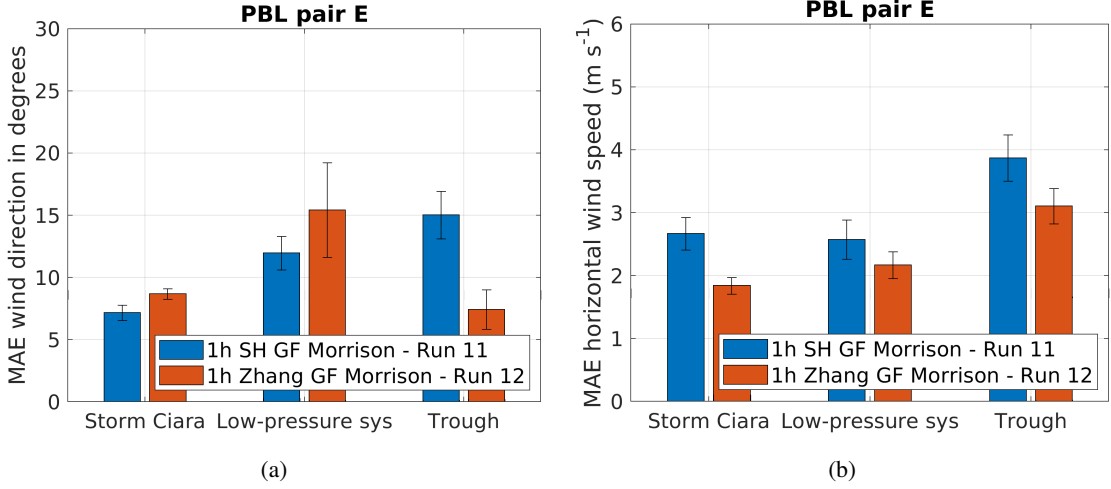

**Figure 12.** Performance evaluation for simulation pair E considering a change in PBL scheme, as described in Table 2. Error bars indicate one standard error of the sample mean. (a) MAE comparison for wind direction. (b) MAE comparison for wind speed.

The current results indicate a high sensitivity in wind direction and wind speed relative to the choice in PBL scheme. A single PBL scheme is not found to outperform the others for all three case studies. The simulation results indicate a possible dependency of PBL schemes to cumulus and microphysics parameterizations, which has also been reported in literature by Hong and Dudhia (2012), Choi and Han (2020), and Chen et al. (2021).

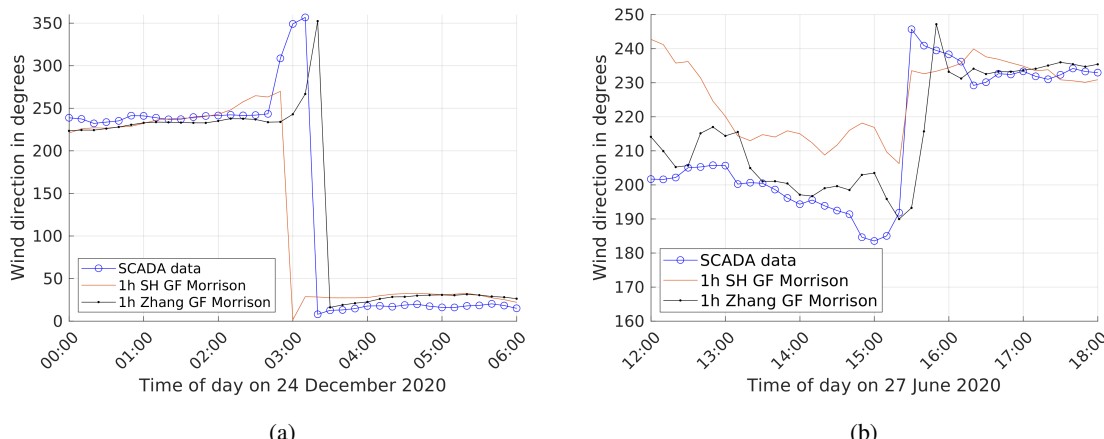

**Figure 13.** Comparison of timeseries considering a change in PBL scheme for simulation pair E. (a) Wind direction plots for low-pressure system case. (b) Wind direction plots for trough passage case.

## 4.6 Cumulus and Microphysics: Simulation pairs F, G, H, I, and J

This section presents and discusses the results for cumulus simulation pairs F, G, and H, as well as microphysics simulation pairs I and J. As these two types of physics schemes both relate to the modeling of precipitation in WRF, they are considered

jointly in this section. A set of eight WRF simulations is considered, covering a combination of four cumulus schemes (KF, GD-3D, msKF, and GF; the latter two being scale-aware) and three microphysics schemes (WSM5, Thompson, and Morrison; in increasing order of modeling complexity).

Figure 14 depicts simulation results for pair F, which uses the SH PBL scheme coupled with the WSM5 microphysics parameterization while varying the cumulus scheme between KF and msKF. Figure 14 presents the results for MAE of wind

direction and wind speed for all three test cases. Starting with wind direction (Fig. 14a), the msKF simulation produces better results for the case of Storm Ciara. However, for the low-pressure system case, KF produces lower MAE which lies within the standard error bars, disallowing statistically significant conclusions. Similarly, for the trough passage, the msKF scheme results in a negligible reduction in MAE. Therefore, conclusions can only be drawn in the case of Storm Ciara, indicating better performance with the msKF scheme on wind direction. Focusing on wind speed (Fig. 14b), again better results for msKF

cumulus scheme are observed for the case of Storm Ciara. For the low-pressure system case both schemes produce similar MAE values. For the trough passage, KF performs better than msKF, reversing the trend found for wind direction. Summarizing, for simulation pair F, the case of Storm Ciara is more accurately predicted by the msKF cumulus scheme in comparison to the KF scheme. For the low-pressure system and trough passage cases, comparative results are inconclusive.

The results for simulation pair G are presented in Fig. 15. Pair G applies the SH PBL scheme coupled with the Thompson

microphysics parameterization, while varying the cumulus scheme between KF and msKF. Overall, the current combination of physics schemes produce similar MAE on wind direction and wind speed compared to simulation pair F. For Storm Ciara, msKF produces lower MAE in wind direction and wind speed. For the low-pressure system, no clear trend in performance is

observed. For the trough passage, lower wind direction MAE is observed for msKF and lower wind speed MAE is observed for KF. Consolidating results for simulation pairs F and G, for the case of Storm Ciara, a clear improvement is observed when using the scale-aware msKF cumulus scheme, however no conclusive statements can be made for the low-pressure system and trough passage cases.

Figure 16 shows results for simulation pair H, which applies the SH PBL scheme coupled with the Morrison microphysics, while varying the cumulus scheme between msKF, GD-3D, and GF. Starting with wind direction (Fig. 16a), Storm Ciara is much better captured by the GD-3D cumulus scheme. For the low-pressure system, no such trend in better performance is observed. For the trough passage, msKF outperforms both GD-3D and GF schemes. Considering wind speed (Fig. 16b), msKF performs better than GD-3D and GF for Storm Ciara. For the low-pressure system, GD-3D is found to be the best performer. For the trough passage, msKF and GF both perform better than GD-3D. However, no distinction is found between msKF and GF schemes. Summarizing, the overall inferences for the simulation pair H are found to be statistically inconclusive and highly sensitive, thus one cannot conclude that a specific cumulus parameterization systematically outperforms the others.

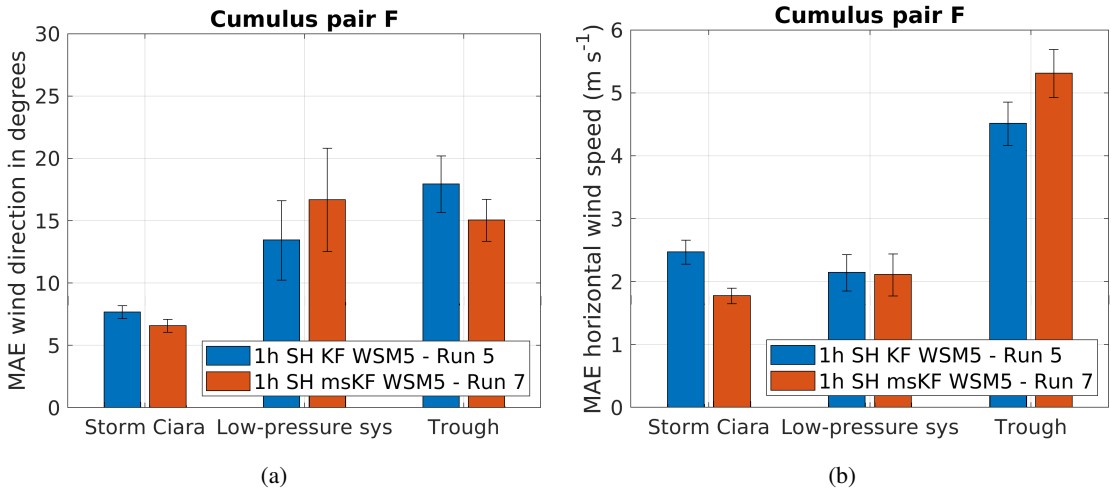

**Figure 14.** Performance evaluation for simulation pair F considering a change in cumulus scheme, as described in Table 2. Error bars indicate one standard error of the sample mean. (a) MAE comparison for wind direction. (b) MAE comparison for wind speed.

Simulation pairs I and J focus on the impact of microphysics schemes. Starting with pair I, results are presented in Fig. 17. Pair I uses scale-aware SH PBL coupled with non-scale-aware KF cumulus schemes and varies the microphysics scheme between WSM5 and Thompson. Overall, no distinction in better performance by either microphysics scheme considering MAE of wind direction and wind speed is observed (Fig. 17a & 17b). For Storm Ciara and the trough passage EWEs, wind speed and wind direction MAE are found to be insensitive to the variation in microphysics schemes. However, this trend is not clear for the case of the low-pressure system, where the Thompson scheme is found to produce different MAE of wind direction, with significantly larger error margins (Fig. 17a & 17b). However, when comparing the combination of cumulus schemes to microphysics setups, more specifically, the combination of WSM5 + KF to Thompson + KF/msKF (Fig. 14a and Fig. 15a),

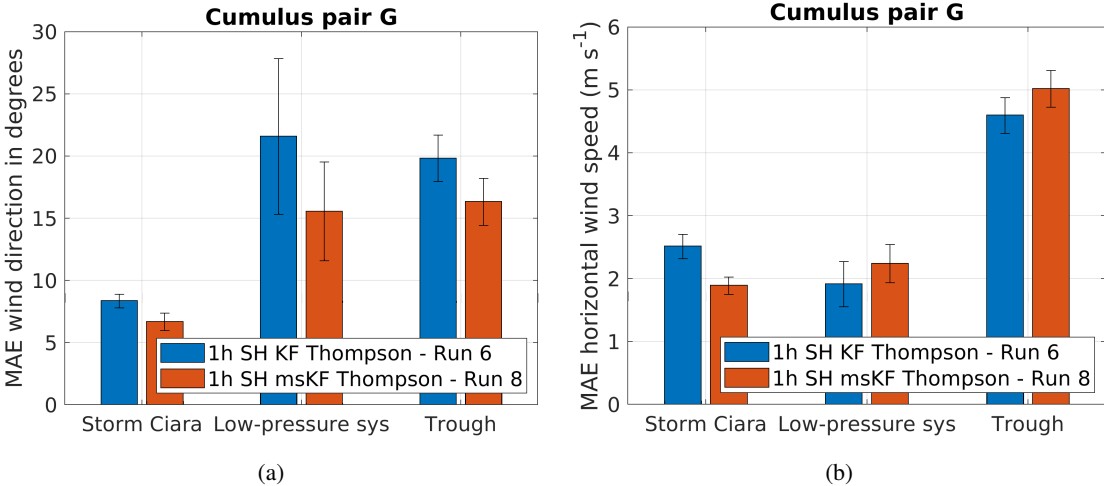

**Figure 15.** Performance evaluation for simulation pair G considering a change in cumulus scheme, as described in Table 2. Error bars indicate one standard error of the sample mean. (a) MAE comparison for wind direction. (b) MAE comparison for wind speed.

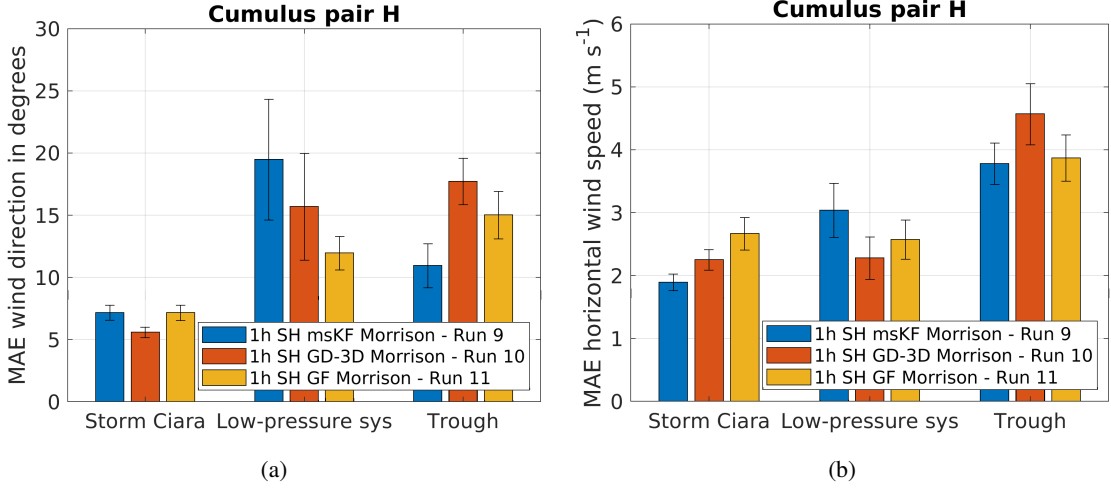

**Figure 16.** Performance evaluation for simulation pair G considering a change in cumulus scheme, as described in Table 2. Error bars indicate one standard error of the sample mean. (a) MAE comparison for wind direction. (b) MAE comparison for wind speed.

lower MAE of wind direction are observed for Thompson microphysics when combined with the scale-aware msKF cumulus scheme. However, this trend is reversed for WSM5 + msKF cumulus schemes. This appears to highlight case-to-case variability
in skill and the importance of the combination of cumulus and microphysics schemes.

Results for simulation pair J, which applies SH PBL with msKF cumulus schemes and varies WSM5, Thompson and Morrison microphysics schemes, are presented in Fig. 18. Considering MAE of wind direction (Fig. 18a), for the case of Storm Ciara and low-pressure system, a clear distinction in the performance of different microphysics schemes is not observed. For

the case of trough passage, Morrison microphysics perform better by comparison. For wind speed MAE (Fig. 18b), Storm Ciara
shows similar results as for wind direction. For the low-pressure system, WSM5 and Thompson produce better wind speed than
Morrison. For the trough passage, Morrison outperforms WSM5 and Thompson microphysics. Overall, for simulation pair J,
Storm Ciara shows an insensitivity to the variation of microphysics schemes. For the low-pressure system, no clear trend in
better performance is observed, whereas a clear advantage in using the more complex Morrison scheme is observed in the
trough passage case.

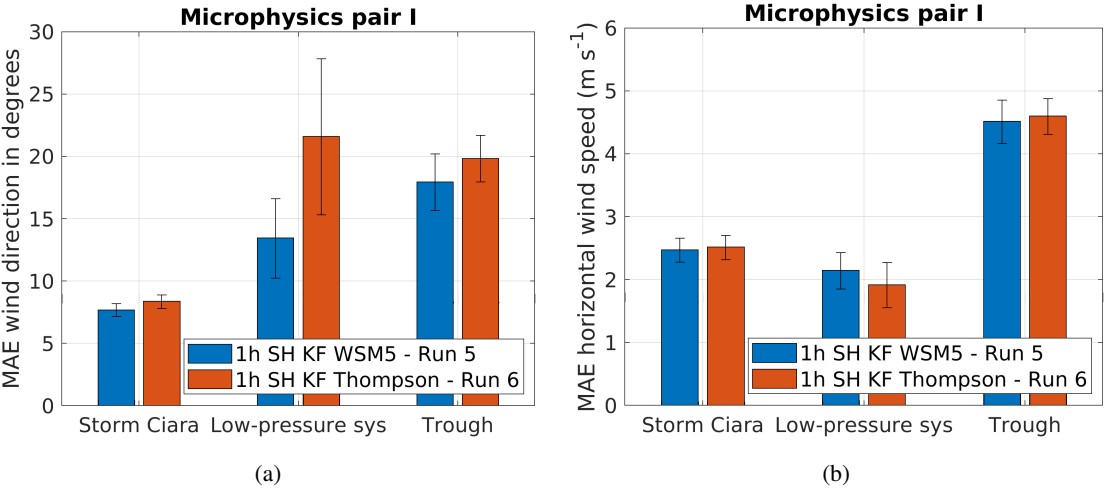

**Figure 17.** Performance evaluation for simulation pair I considering change in microphysics schemes, as described in Table 2. Error bars
indicate one standard error of the sample mean. (a) MAE comparison for wind direction. (b) MAE comparison for wind speed.

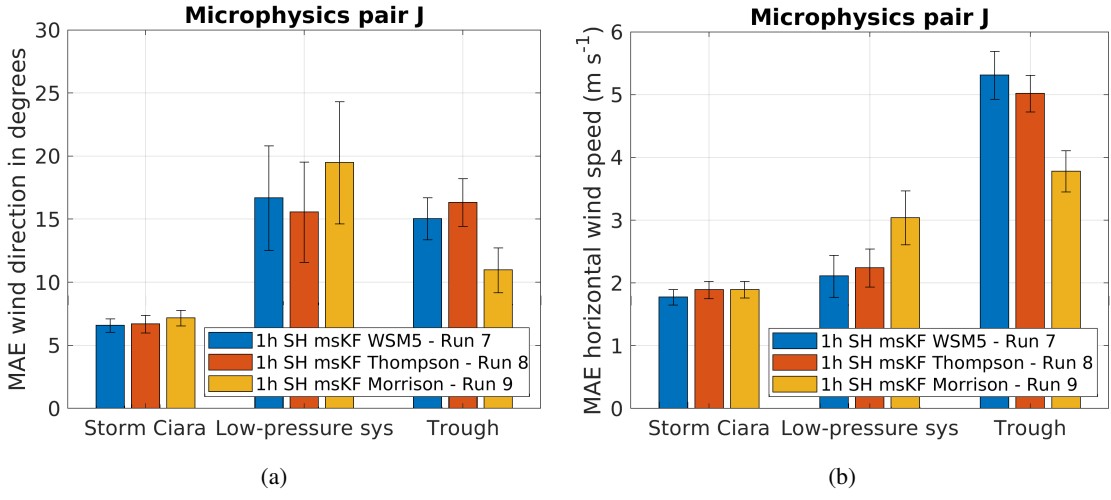

**Figure 18.** Performance evaluation for simulation pair J considering change in microphysics schemes, as described in Table 2. Error bars
indicate one standard error of the sample mean. (a) MAE comparison for wind direction. (b) MAE comparison for wind speed.

## 4.7 Discussion

The previous sections investigated the individual influence of varying a single physics parameterization in the modeling chain on the accuracy of the match between WRF simulation results and observational data when subject to three EWE case studies. A clear trend in improved performance with higher model complexity common to all case studies is not found.

When looking at the update interval of LBCs, the qualitative differences in hourly and 3-hourly update intervals are found to be marginal. The quantitative indicators show a unanimous improvement for the case of Storm Ciara (see Fig. 8 & 9). However, for the low-pressure system and the trough passage this distinction is not so evident.

The variation in PBL scheme results in highly sensitive metrics for all three case studies. For the case of Storm Ciara, a clear advantage in using scale-aware SH PBL in comparison to non-scale-aware MYNN PBL is observed (see Figs. 10 & 11). However, this trend is not evident for the case of the low-pressure system and the trough passage. When comparing scale-aware schemes of different fidelity, more specifically Zhang PBL and SH PBL, a promising trend is observed in which the higher complexity Zhang PBL leads to a better match with SCADA data for the trough passage case (Fig. 13). However, this trend is not found for Storm Ciara and the low-pressure system. Furthermore, considering wind speeds, the Zhang PBL run is either the best setup (cold front) or results in MAE very close to the best setup (Storm Ciara and low-pressure system). In contrast, the Zhang PBL setup results in higher wind direction errors for Storm Ciara and the low-pressure system cases.

Regarding the cumulus and microphysics simulation pairs, the combination of cumulus and microphysics is observed to have more impact on MAE of wind direction in comparison to variation in stand-alone cumulus or microphysics schemes. This is highlighted by Storm Ciara and the low-pressure system, where the change in microphysics schemes in combination with msKF cumulus results in marginal changes in MAE (see Fig. 18). However, when comparing the combinations of lower-order microphysics with scale-aware and non-scale-aware cumulus schemes for wind direction, i.e., WSM5/Thompson + KF/msKF, results indicate an overall reduction in MAE for Thompson + msKF (see Fig. 14a & 15a). These results potentially indicates scale-aware cumulus schemes to be more compatible with higher-order microphysics schemes. The performance of cumulus and microphysics schemes is found to be strongly dependent on the type of weather phenomenon.

Precipitation results were qualitatively compared to each other and to the radar images from RMI-B (Fig. 1). It was found that all WRF simulation precipitation results are highly sensitive to the combination of physics schemes and type of EWE, yet no conclusions on modeling fidelity could be drawn from this analysis, and hence further discussion is omitted here. As an example, a sample of results for the case of Storm Ciara is presented in Fig. 19, illustrating the completely different precipitation fields produced by different model setups. A direct quantitative comparison of simulated reflectivity and observed raw radar fields using, e.g., tools for comparing gridded observations such as MODE (Newman et al., 2022), is impeded by the lack of filtering and post-processing information on the raw radar observation data. Therefore, a quantitative assessment of precipitation modeling is out of scope of the current paper and left for future work.

Finally, the ensemble average (as defined in Sect. 3.3) is observed to rank very similar to the best-case model setup (see Table 3 & 4) for the cases of Storm Ciara and the low-pressure system. However, the fast changes in wind direction are dampened

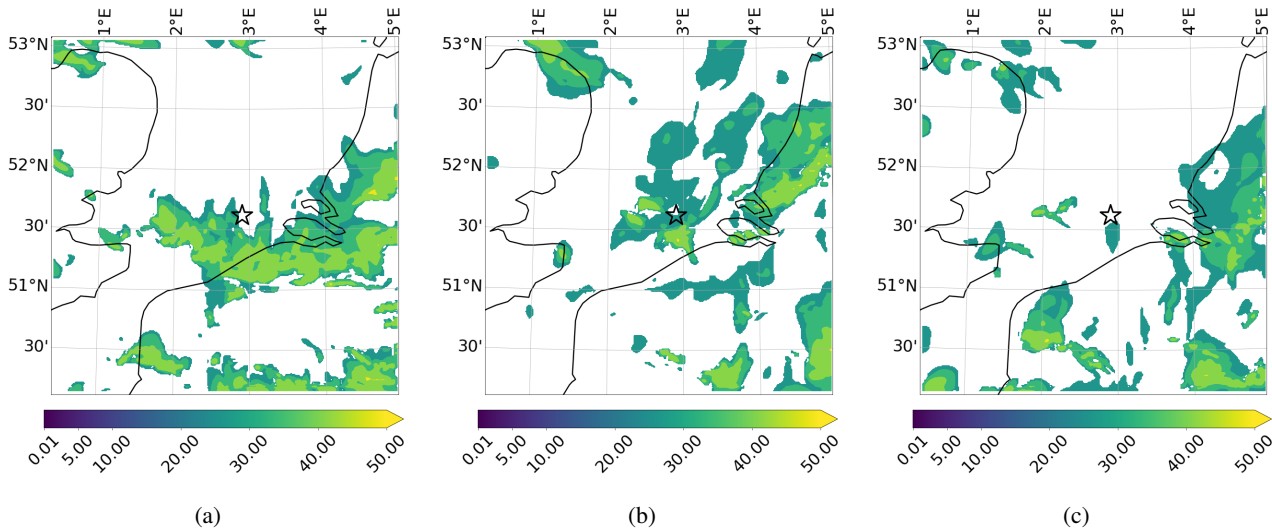

**Figure 19.** Contours of WRF precipitation rate in mm h$^{-1}$ for the case of Storm Ciara on 10 February 2020 at 04:40 UTC. The plots are presented for cumulus simulation pair H for domain d04. The star in the plots represents the offshore wind farm of interest. (a) Simulation run 9: 1h SH msKF Morrison, on 10 February 2020 at 04:40 UTC (b) Simulation run 10: 1h SH GD-3D Morrison, on 10 February 2020 at 04:40 UTC (c) Simulation run 11: 1h SH GF Morrison, on 10 February 2020 at 04:40 UTC.

by the ensemble averaging (see Fig. 5 & 6). For the trough passage, the ensemble averaging performs poorly compared to the best-case setup by a significant margin (Table 5), caused by a persistent offset by all but the best-case setup.

## 5 Conclusions and recommendations

The complexity in determining an optimal combination of physics setup for the operational use of the WRF model in the frame of wind energy applications over the Belgian North Sea is analyzed in this study. A multi-event sensitivity analysis for WRF NWP model is performed considering three extreme weather events: Storm Ciara on 10 February 2020, a low-pressure system on 24 December 2020 and a trough passage on 27 June 2020. These events have been identified to be potentially harmful for the operation of offshore wind farms. The events resulted in fast changes in wind direction leading to severe yaw misalignment of the turbines, with the potential to result in significant off-design turbine load cases and variability in power production. This sensitivity analysis utilizes operational wind farm data (SCADA) for evaluating WRF simulated wind direction and wind speed results. Qualitatively, precipitation results are found to be highly sensitive to model setup and type of EWE. No clear tendency towards better accuracy with increased complexity of parameterizations is found. This sensitivity study analyses the impact of the update interval of the LBCs and sub-grid scale modeling techniques used for the PBL, cumulus, and microphysics parameterizations.

The results of this sensitivity analysis indicate WRF simulations to be highly sensitive to the type of event and the combination of physics parameterizations. Starting with the variation in update interval of LBCs, overall better performance for hourly

update interval of LBCs is observed for the case of Storm Ciara. However, for the low-pressure system and trough passage cases no such trend is observed. In general, WRF simulations comprising scale-aware PBL physics schemes appear to perform better in comparison to non-scale-aware physics schemes, as the best-case setups for all three events feature scale-aware PBL schemes. Concerning cumulus and microphysics parameterizations, the suitable combination of cumulus and microphysics is observed to be highly dependent and sensitive to the type of weather phenomenon. The combination of schemes is observed to have more impact than a stand-alone variation for either of these events.

Overall, in view of modeling local wind direction and wind speed at the location of the farms, three independent best-case setups are identified for the three case studies. A single best WRF model setup for both wind direction and wind speed for all three case studies is not found. The results indicate little consistency across the three EWEs for different parameterizations. For the case of Storm Ciara, the best-case setup is identified to combine the scale-aware SH PBL scheme coupled with the scale-aware msKF cumulus parameterization and five-class single moment WSM5 microphysics. For the low-pressure system, the best-case setup combines, scale-aware SH PBL, non-scale-aware KF cumulus, and WSM5 microphysics schemes. For the trough passage, the best-case setup is identified to combine scale-aware Zhang 3D PBL, scale-aware GF cumulus, and six-class double moment Morrison microphysics schemes. The best-case setups for all cases utilize hourly reanalysis dataset as the LBCs and scale-aware PBL schemes. Overall, the inconsistency across different EWEs found in the current work suggests that a general best model configuration for the Belgian North Sea does not exist, and that best practices are highly dependent on the weather regime under consideration. However, it is important to note that this conclusion is based on a limited sample of EWEs over a single observation point at the offshore wind farm. To further justify and generalize this conclusion, a much larger sample considering more weather events, more observations, and more model configurations is warranted.

An interesting area of further research would be to perform similar sensitivity studies at finer sub-kilometer resolutions including recent advancements such as 3D scale-aware PBL schemes (Zhang et al., 2018; Senel et al., 2020). Furthermore, expanding the sensitivity analysis to include events such as a dunkelflaute (Li et al., 2021) and wind ramps (Gallego-Castillo et al., 2015) will allow a broader assessment of EWE modeling relevant to wind energy. Also, a quantitative assessment of ground-level precipitation modeling with local precipitation measurements from disdrometers and tipping buckets is of general interest to assess, e.g., the risk of leading edge erosion of wind turbine blades (Law and Koutsos, 2020).

*Acknowledgements.* This work has received funding from the Flemish Government through the Agency for Innovation and Entrepreneurship (Vlaams Agentschap Innoveren en Ondernemen, VLAIO) through the cSBO project SeaFD in the context of the MaDurOS program for Material Durability for Off-Shore of the cluster on Strategic Initiative for Materials in Flanders (SIM). Furthermore, the authors acknowledge VLAIO funding through the RAINBOW project in the context of SIM and the Blue Cluster. The authors thank dr. Laurent Delobbe from the Royal Meteorological Institute of Belgium for providing radar reflectivity data from the dual-polarization C-band radar, located in Jabbeke at the Belgian coast.

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
