# Peer review of "Sensitivity analysis of mesoscale simulations to physics parameterizations over the Belgian North Sea using WRF-ARW"

_Wind Energy Science, 2021_

## Author Comment (AC1)

**Author comments in reply to anonymous referee #1**

We thank anonymous referee #1 for their useful comments and the time invested in reviewing our manuscript. We very much appreciate the critical and thorough review of our work. We have addressed each of the referee comments as detailed point by point below, which we believe has significantly improved the quality of the manuscript. We hope our revised manuscript can be accepted for publication.

Adithya Vemuri, on behalf of all co-authors

**Overarching comments**

To fully address the major comment#2, we have added 2 additional extreme weather events to this sensitivity analysis. In the revised manuscript all extreme events observe fast changes in wind direction accompanied by severe yaw misalignment leading to potentially harmful conditions for wind farm operation. Furthermore, we have emphasized statistical significance of the differences between simulation setups and have therefore refocused the analysis on the quantitative MAE analysis of wind speed and wind direction. As a result, the revised manuscript observes significant changes to its structure, simulation setups, simulations pairs and the evaluation methodology.

Further, we mention that during the revision, we have checked and refined our post-processing tools, which led to some minor changes in metric values and time-series plots for the case of Storm Ciara.

**Major comments**

**Comment 1:** *A thorough technical English edit is required. There are numerous issues throughout the manuscript, most of which are specifically mentioned in the Minor Comments from the Abstract through Section 2 as examples of the issues that need fixing. I only mentioned a couple items from Sections 3–5.*

We thank the referee for pointing this out. We have spent significant attention to improving the overall quality of the English language with a native English speaker, both for the specific points raised by the referee, and for the manuscript in general. This is further detailed in our reply to the minor comments below.

**Comment 2**: *Lines 95–98 provide the key to defining the niche that this article aims to fill. I think you need to do a better job honing in and repeatedly showing how your work fills that niche. I also think that validating against data at only a single point for a single storm is inadequate to fill that niche. That is really the biggest fundamental issue I have with this manuscript, and why I gave it a Reject instead of a Major Revision. A single case study can have value if you do more than validate against observations only at a single point, while validating at a single point can have some value if you evaluate multiple cases. Are there any other buoys or towers that are available for offshore wind validation? I am aware of FINO1 in the North Sea region, but I believe it is outside your d04, unfortunately.*

We thank the referee for this valid comment. We address it in 2 ways as shown below. Modifications to the revised manuscript are highlighted in blue.

- Lines 95-98 indeed sparingly address one of the niche areas this paper aims to fill, on the applicability of Shin-Hong PBL scheme to coastal environments. Lines 38-45 establish the fundamental motivation and novelty behind this study, i.e., to perform a sensitivity analysis on

WRF physics parameterizations for the Belgian offshore wind farms from an operational SCADA point of view.

In the introduction of the revised manuscript, we emphasize the motivation in a better way:

Line 41: Sensitivity analyses are typically conducted to identify the optimal combination of physics schemes for a specific location (see, e.g., Efstathiou et al. 2013; Santos-Alamillos et al. 2013; Kala et al. 2015). This type of investigation has not been performed for the Belgian North Sea. Furthermore, to the authors' best knowledge, no previous studies have looked at potentially harmful EWE from a wind farm perspective as experienced by the machines themselves. Therefore, this sensitivity analysis aims to address this gap in research. The analysis presented in this paper assesses the impact of a wide range of physics parameterizations for PBL, cumulus and microphysics, and length of the update interval of LBC (lateral boundary conditions) on the simulated wind direction and speed.

This is again highlighted in the methodology section:

Line 170: This evaluation uses operational wind farm SCADA data for its quantitative analysis of wind direction and speed. Additionally, radar data from RMI-B allows for a qualitative perspective on precipitation. By combining these observational datasets, the premise of this study provides a unique opportunity to investigate EWE as experienced by an offshore wind farm to determine suitable WRF setups in the specific context of wind energy applications.

- As rightfully mentioned, d04 in our simulations does not encompass many public-domain observational datasets suitable for offshore wind validation. However, we agree that extending the analysis to multiple cases significantly enhances the quality of our manuscript. Therefore, to address the second part of this comment we include 2 additional extreme weather events that observe fast changes in wind direction accompanied by severe yaw misalignment. Furthermore, we have also opted for including statistical uncertainties in a more objective and quantitative analysis to assess which trends persist with statistical significance over different events. In this regard, we have refocused our analysis on quantitative comparison of the wind direction and speed MAE. We have removed quantitative discussions on precipitation (see also comment 3) and Kantorovich distances (for which we cannot define any error bars based on the available data). Furthermore, we have removed the somewhat subjective domain configuration sensitivity (see also comment 4d) in favor of an additional case with the 3D Zhang PBL scheme. As a result, we still define 12 runs per event (shown in table below), or a total of 36 WRF configurations for the 3 events combined.

| Simulation run# | ERA5 LBC updates | PBL scheme | Cumulus scheme | Microphysics scheme | Update interval pairs | PBL pairs | Cumulus pairs | Microphysics pairs |
|---|---|---|---|---|---|---|---|---|
| 1 | 3 h | MYNN | msKF | WSM5 | A | | | |
| 2 | 3 h | Shin | KF | WSM5 | B | | | |
| 3 | 1 h | MYN | KF | Thompson | | C | | |
| 4 | 1 h | MYNN | msKF | WSM5 | A | D | | |
| 5 | 1 h | Shin | KF | WSM5 | B | | F | I |
| 6 | 1 h | Shin | KF | Thompson | | C | G | |
| 7 | 1 h | Shin | msKF | WSM5 | | D | F | J |
| 8 | 1 h | Shin | msKF | Thompson | | | G | |
| 9 | 1 h | Shin | msKF | Morrison | | | H | |
| 10 | 1 h | Shin | GD-3D | Morrison | | | | |

| 11 | 1 h | Shin | GF | Morrison | | | | |
|----|-----|------|----|----------|---|---|---|---|
| 12 | 1 h | Zhang | GF | Morrison | | E | | |
| 13 | Ensemble average | | | | | | | |

**Comment 3:** *Validating precipitation at only a single point is of limited use, even over many cases, when your goal is to determine which model configuration gave the most realistic simulation of precipitation. If you want to validate model precipitation or reflectivity, then you should leverage the land-based radar that you do have data from and do object-based validation with MODE for a more comprehensive validation.*

We thank the reviewer for this useful comment. We agree that single-point validation is of limited use, and that a MODE met-plus analysis tool would allow for a more comprehensive quantitative precipitation study. However, we are currently limited by data access to the land-based radar, with access to only the raw reflectivity radar data which, in itself, is not currently publicly available. We also lack the information required to post-process reflectivities into a form that could be quantitatively compared to WRF output, or to extract rainfall at ground level from this data as described, e.g., in Ref [1] below. Therefore, we omit any further quantitative evaluation of modeled precipitation in the revised manuscript. However, since precipitation plays a major role in wind energy and extreme weather events, we briefly mention the sensitivity and variability of precipitation rates produced by the model in the discussion section. Quantitative assessment of precipitation is mentioned as an area of future work in the conclusions section.

[1] Goudenhoofdt et al. 2016, Generation and Verification of Rainfall Estimates from 10-Yr Volumetric Weather Radar Measurements in: Journal of Hydrometeorology Volume 17 Issue 4 (2016)

The following lines have been included in the discussion section:

> Line 401: A qualitative perspective on precipitation indicates all WRF simulations to be highly sensitive to the combination of physics schemes and type of EWE. The qualitative analysis yielded little to no conclusions on the precipitation modelling fidelity of the considered WRF physics setups in this study. As an example, the results for the case of Storm Ciara are presented in Fig. 19 (in the revised manuscript). A direct quantitative comparison of simulated reflectivity and observed raw radar fields using, e.g., tools for comparing gridded observations such as MODE [2] is impeded by the lack of filtering and post-processing information on the latter raw data. Therefore, a quantitative assessment of precipitation modeling is out of scope of the current paper and left for future work.

> [2] Newman, K., J. Opatz, T., Jensen, J., Prestopnik, H., Soh, L., Goodrich, B., Brown, R. B., and Gotway, J. H.: MET-MODE, in: The MET Version 10.1.0 User's Guide, DTC, 2022

Furthermore, we explicitly mention the radar data is not publicly available in the description of the events section:

> Line 137: Three case studies are considered in this sensitivity analysis, namely, Storm Ciara on 10 February 2020, a low-pressure system on 24 December 2020, and a trough passage on 27 June 2020. The radar data presented therein is not publicly available, but was retrieved through a bilateral agreement with the Royal Meteorological Institute of Belgium (RMI-B). A brief synopsis of these events is presented in the following sub-sections.

**Comment 4:** *Updates on figures*

a) *Fig. 1 (several of these comments also apply to Figs. 6, 8, 10, and 11): It is customary to plot coastlines or national borders in black or gray on most maps. Using a color from your colorbar (red) is simply confusing. Also, your filled contour colors do not match the colorbar. The thin ribbons of darker colors around fields of pastel colors also make this figure difficult to interpret with any confidence. Additionally, in this figure the colorbar label says "Precipitation" but has units of mm/h. Precipitation would have units of mm, but mm/h are units for precipitation rate. The caption also states that the figure is depicting radar reflectivity, which again is not quite the same thing as precipitation rate, though they of course are related to one another. The caption also states that the observed radar reflectivity (or really, radar-derived precipitation rate) is valid at 04:00, but line 137 says it is valid at 04:40. Which is it?*

• The precipitation plots are changed to address these comments. An example plot format for the change in cumulus parametrizations is presented below. Thank you for pointing out discrepancies in the definition of precipitation units, we have updated the figure caption to reflect the correct unit of precipitation rate as mm/h. The time stamp for the case of Storm Ciara is at 04:40 UTC, also has been updated in the revised manuscript.

[Figure]

Fig. 19 caption: Contours of WRF precipitation rate in mm h$^{-1}$ for the case of Storm Ciara on 10 February 2020 at 04:40 UTC. The plots are presented for cumulus simulation pair H for domain d04.

b) *Figs. 3 and 4: You should have a thin gray line in your legend if it is in your plot. In the x-axis label (also in Figs. 5, 7, and 9), also use a date format like 10 Feb 2020. 10/2/2020 will easily confuse American readers into thinking the date is 2 Oct 2020.*

c) *Fig. 5: The orange dashed line is quite faint and difficult to see.*

• We agree the date format can be confusing to readers of different region. Therefore, we have adopted the unambiguous date format as, e.g., 10 February 2020. The timeseries plots have also been updated to address the formatting comments while keeping in mind the color-blind readability. Example plots are presented below.

[Figure]

[Figure]

d) *Fig. 6 (most of these comments also apply to Figs. 8, 10, and 11): First, calling these Domain 1 and Domain 2 is misleading. These are really Domain Configurations 1 and 2; both these domain configurations have domains 1–4, so when you say domain 1 or domain 2, my mind automatically thinks of the outermost two WRF domains. Second, the radar contour lines all look the same color, which seems like a mistake. Third, restrict your WRF filled contour range to the equivalent of the radar derived precipitation rate, or at least restrict the lower bound cutoff to something like 0.01 mm/h. Is it really raining at lower values than that, anyway? Values below your lower bound should be transparent/not plotted. That will also solve the undesirable issue of the entire domain being filled with a dark blue that makes other features difficult to discern while also being meaningless. Fourth, state in the caption what the star is.*

- Although only a single domain configuration is considered in the revised manuscript, the comment on addressing the right terminology for domain configuration is still relevant to the revised manuscript. We update the captions of WRF domain configuration and nested domains throughout the paper.

  The following changes are made to the plot caption of the WRF domain configuration:

  Fig. 4: WRF and WRF Post-processing System (WPS) nested domain configuration (1-way nesting) considered common to all simulation runs in this study.

- In view of the reply to comment 3, we omit the radar contours from WRF precipitation rate plots, the updated plots are presented above (Fig. 19).

- Indeed, the precipitation rate from WRF simulations that was being plotted was lower than 0.01mm/h. Therefore, we update these plots considering precipitation rate values lower than 0.01mm/h to 0. Furthermore, we have amended the useful suggestions in the revised figures. These updated plots are shown above in the reply to Comment 4a.

**Comment 5:** *To be completely honest, changes in wind direction of 40° do not seem like a huge shift—it is not even half of a quadrant. If 40° is a hugely consequential shift that wind farm operators need to be quite concerned about, then it would be helpful to provide some justification.*

- The events featured in this study are selected based on specific alarms raised by individual turbines during fast changes in wind direction and severe yaw misalignment. However, bounds on such yaw misalignment and thresholds for raising alarms are specific to turbine models and wind farm operators. As this data is highly confidential, we cannot further elaborate in detail on this in the manuscript. However, we do understand the comment by the referee, and address this by elaborating on the methodology used for the identification of the extreme weather events.

    The following lines are added to the methodology section:

    Line 128: The selection of the events in this study is motivated by the occurrence of fast changes in wind direction accompanied by severe yaw misalignment leading to significant power loss as observed by a Belgian offshore wind farm in the North Sea. The methodology utilized to identify these events modifies the approach defined by Hannesdóttir and Kelly (2019) to include yaw misalignment. The wavelet analysis considers a minimum threshold to identify anomalous changes in wind direction accompanied by severe yaw misalignment experienced by several wind turbines. Severe yaw misalignment potentially has adverse effects on the operational lifetime and fatigue loading of a wind turbine (Wan et al., 2015; Bakhshi and Sandborn, 2016; Laino and Hansen, 1998; Damiani et al., 2018), highlighting its importance and relevance in this study. The SCADA analysis for the identification of these events includes confidential error codes and data that are protected under a non-disclosure agreement, therefore no further details can be provided herein.

**Comment 6:** *Table 2: I suggest either reordering pairs A–K based on the order they are discussed in Sections 4.1–4.4, reordering Sections 4.1–4.4, or both (my preference). It makes more sense to me to look first at the size of the domain and the lateral boundary condition temporal frequency, before then comparing different physics schemes. Also, in Table 2 ensure that cell borders are turned on to separate the different experiment pairs in the cumulus pair column. Additionally, in the section titles for Sections 4.1–4.4, it would be helpful to include the experiment pair letters.*

- We thank the referee for this insight and realize this could be confusing to the reader. The revised paper structure is re-organized to make the readability easier. The PDF was rendering at lower resolution, making the cell borders invisible in some places. This is improved in the revised manuscript, an example table for Storm Ciara is presented below (Table 3 in the revised manuscript).

**Comment 7:** *Table 7: In the Average NED column for rows 10 and 11, you have 1.10 in green, 1.111 in yellow, and 1.32 in red. The values 1.10 and 1.111 are so close that it is misleading to make them such different colors. Is the difference between 1.10 and 1.111 in this metric even meaningful? What would be a meaningful difference in NED or Kantorovich distance? In any case, in Tables 4–7, I really think you would be better off keeping the color scale and ranges from Table 3.*

- We agree with this comment and have categorized the values into 5 colors to avoid confusion.

    Following lines were added to the methodology section to highlight the same:

Line 229: Cells are colored based on a set of 5 categories between red and green. Categories are defined to cover 20% of the range between smallest and largest values for the considered metric. In this way, results are categorized into best (green, with errors in the 20% lowest range), good (light green), average (yellow), poor (light red), and worst (dark red).

The performance evaluation tables are updated to reflect the same, as an example the performance evaluation table (Table 3 in the revised manuscript) for the case of Storm Ciara is presented here.

| Simulation run# | ERA LBC updates | PBL scheme | Cumulus scheme | Microphysics scheme | Wind direction MAE (degrees) | Wind speed MAE (ms$^{-1}$) | NED (-) |
|---|---|---|---|---|---|---|---|
| 1 | 3 h | MYNN | msKF | WSM5 | 10.46 | 3.88 | 2.08 |
| 2 | 3 h | Shin | KF | WSM5 | 8.48 | 2.57 | 1.51 |
| 3 | 1 h | MYNN | KF | Thompson | 9.26 | 2.72 | 1.63 |
| 4 | 1 h | MYNN | msKF | WSM5 | 8.61 | 2.54 | 1.51 |
| 5 | 1 h | Shin | KF | WSM5 | 7.68 | 2.47 | 1.41 |
| 6 | 1 h | Shin | KF | Thompson | 8.37 | 2.51 | 1.48 |
| 7 | 1 h | Shin | msKF | WSM5 | 6.59 | 1.78 | 1.11 |
| 8 | 1 h | Shin | msKF | Thompson | 6.69 | 1.89 | 1.15 |
| 9 | 1 h | Shin | msKF | Morrison | 7.17 | 1.89 | 1.20 |
| 10 | 1 h | Shin | GD-3D | Morrison | 5.59 | 2.25 | 1.17 |
| 11 | 1 h | Shin | GF | Morrison | 7.17 | 2.67 | 1.43 |
| 12 | 1 h | Zhang | GF | Morrison | 8.69 | 1.84 | 1.34 |
| 13 | Ensemble average | | | | 5.88 | 2.04 | 1.12 |

**Comment 8:** *Lines 325–327: First, change "ensemble" to "ensemble mean" or "ensemble average". Second (and more importantly), there are many papers and books that explain why the ensemble mean usually outperforms individual ensemble members (e.g., Wilks 2019, https://www.elsevier.com/books/statistical-methods-in-the-atmosphericsciences/ wilks/978-0-12-815823-4). It would be worthwhile to engage with some of that literature here, especially since your findings of the ensemble mean not being the best are contrary to what was expected. Do you have any insights as to why the ensemble mean performs comparatively poorly in the Kantorovich distance for wind speed and wind direction? This appears to be why the NED is not the best for the ensemble mean for the wind variables. Perhaps this is a side effect of the randomness introduced by having a sample size of only one event validated at only one point?*

- We thank the referee for pointing out a possible source of confusion regarding the ensemble average. We have updated the performance tables and timeseries plots (as depicted above) to state ensemble average instead of just ensemble. The definition of ensemble average in this study differs from a more traditional ensemble forecast, as defined in Wilks 2019, in that our ensemble average only includes the variability in physics scheme and the change in temporal resolution of boundary conditions, and hence does not reflect any uncertainties on initial conditions. We reflect on the literature and differentiate the ensemble average considered in this study in the methodology section.

Following lines have been added to the methodology section to highlight the above point:

Line 217: This study also evaluates the performance of an ensemble average compared to single deterministic simulation runs. The ensemble average is defined as the mean of all simulation runs considered for a given case study. In this study, ensemble members are initialized with identical initial conditions from ERA5 reanalysis. Subsequently, variability in the ensemble average is only caused by the variation in update interval of LBC and physics parameterizations. Therein, the current definition of ensemble average differs from traditional ensemble forecasts, where variations in initial conditions are also considered, see, e.g., Wilks (2019).

- As mentioned above, with the addition of 2 more events to the revised manuscript, we have included a statistical comparison to quantify uncertainties on the MAE estimates and assess which trends persist with statistical significance over different events. An example plot (Figure 12b in the revised manuscript) is presented below indicating the change in performance evaluation for simulation pairs.

[Figure]

**Minor comments**

We thank the referee for their critical review on the manuscript's written English. The typos, comma style, abbreviation definitions and technical detail have been corrected in the revised manuscript. In addition to the specific points raised by the referee, we have thoroughly revised the use of English throughout the manuscript with the help of a native English speaker.

We have amended all the textual changes suggested by the referee. For brevity, we only mention the ones that require further explanation.

- *Throughout: Whether you use Oxford commas or not, the journal guidelines state that you need to be consistent, but there is not consistency of usage in your article.*

We have consistently made use of Oxford commas in the revised manuscript.

- *Throughout: Provide the time zone for all times in this article. I presume your times are in UTC, but it is never stated.*

Indeed, time zones are UTC, we mention this throughout the revised manuscript.

- *Line 7: It would be nice to mention in the abstract when Storm Ciara occurred.*

The dates of the events considered are now included in the abstract.

- *Line 30: Change "open-source" to "public domain". Also, since you use WRF v4.2, you should instead cite the WRF v4 technical note (Skamarock et al. 2019). Consider also citing Powers et al. 2017.*

We thank the referee for pointing this out and included both references.

- *Line 111: "concluded in no particular combination of WRF physics" — This is awkwardly worded. Please revise.*

The sentence has been revised to:

Line 114: In contrast, the study by Islam et al. (2015) for the Haiyan tropical cyclone over the west Pacific Ocean did not identify a suitable combination of WRF physics to best reproduce the extreme weather event.

- *Line 114: "time-lapse considered within the diurnal cycle" — This is awkwardly worded. Please revise.*

The sentence has been revised to:

Line 116: … analyses indicating a wide array of possible combinations of physics parameterizations depending on the type of weather phenomenon, the season and the time period considered for simulation within the diurnal cycle.

- *Line 124: "exposed" is an odd word choice here.*

The sentence has been revised to:

Line 125: Lastly, conclusions and future prospects are presented in Sect. 5.

- *Line 132: Please define what you mean by "over the local region". How local? D04 is not very big to begin with.*

Indeed, d04 is too small to capture all affected areas in Belgium by Storm Ciara. The areas affected as reported by RMI-B (RMI - Storm Ciara (meteo.be)) extend far in-land and offshore. The sentence is revised to:

Line 146: … the Royal Meteorological Institute - Belgium (RMI-B) reported wind gusts of up to 115 km h$^{-1}$ in Ostend, located at the Belgian coast, with heavy precipitation accompanied by strong winds and thunderstorms.

- *All throughout Section 4: I think you need to consistently refer to domain configuration 1 and 2, as each domain configuration has domains 1, 2, 3, and 4. If you say domain 1, the reader will think of your outermost WRF domain, which is domain 1 (d01).*

We agree with this comment and have amended it (see also reply to comment 4d)

- *Line 322: This sentence is awkwardly worded.*

The sentence is revised to:

Line 401: A qualitative perspective on precipitation indicates all WRF simulations to be highly sensitive to the combination of physics schemes and type of EWE. The qualitative analysis yielded little to no conclusions on the precipitation modelling fidelity of the considered WRF physics setups in this study.

---

## Author Comment (AC2)

**Author comments in reply to anonymous referee #2**

We thank anonymous referee #2 for their useful comments and the time invested in reviewing our manuscript. We very much appreciate the critical and thorough review of our work. We have addressed each of the referee comments as detailed point by point below, which we believe has significantly improved the quality of the manuscript. We hope our revised manuscript can be accepted for publication.

(Throughout this document, specific modifications to the revised manuscript are shown in blue)

Adithya Vemuri, on behalf of all co-authors

**Overarching comments**

To fully address both of our referee comments, in response to major comment 2 by referee #1, we have included 2 additional extreme weather events to this sensitivity study. Therefore, the revised manuscript observes significant changes to its paper structure, simulation setups, simulation pairs and the evaluation methodology considered.

With the addition of 2 more events to the revised manuscript, we have also opted for including statistical uncertainties in a more objective and quantitative analysis to assess which trends persist with statistical significance over different events. In this regard, we have refocused our analysis on quantitative comparison of the wind speed and direction MAE. We have removed quantitative discussions on precipitation and Kantorovich distances (for which we cannot define any error bars based on the available data). Furthermore, we have removed the somewhat subjective domain configuration sensitivity in favor of an additional case with the 3D Zhang PBL scheme, resulting in a total of 36 WRF simulations for the 3 events combined. An example plot (Figure 12b in the revised manuscript) is presented below indicating the change in performance evaluation for simulation pairs.

Further, we mention that during the revision, we have checked and refined our post-processing tools, which led to some minor changes in metric values and time-series plots for the case of Storm Ciara.

**Specific comments**

**Comment 1:** There are major inconsistencies in the way radar reflectivity and precipitation (rate) are handled in the manuscript. This is especially visible in some of the figures (captions mentions reflectivity

while figure legends talk about precipitation with units of precipitation rates, e.g. Figure 1, 4), but also in the text (e.g. line 188-191). A major revision of all elements of the manuscript is needed by the authors with special attention to a consistent usage of atmospheric properties and units.

• We thank the referee for pointing this out and have corrected for the inconsistencies in all figure and precipitation mentions. An example caption from the revised manuscript is presented below.

Fig. 19 caption: Contours of WRF precipitation rate in mm  $h^{-1}$  for the case of Storm Ciara on 10 February 2020 at 04:40 UTC. The plots are presented for cumulus simulation pair H for domain d04.

**Comment 2:** Throughout the paper, the term "horizontal resolution" has been used to describe the grid spacing of the WRF domains (among others in Table 1, line 145/146 and elsewhere). This is misleading since horizontal (effective) resolution and grid spacing are not equivalent for numerical models like WRF (see e.g. Skamarock 2004 for details). Please replace the term "horizontal resolution" with "grid spacing" where needed.

• Indeed, we agree that grid spacing and horizontal resolution are not equivalent, and have corrected this throughout the revised manuscript, also in Table 1 (presented further in this reply).

**Comment 3:** As mentioned earlier, the descriptions in the methodology section are quite convoluted with model setup, evaluation metrics, introduction and processing of measurements and WRF post-processing all described in the same section. I would suggest to introduce subsections for the model setup (incl. WRF post-processing of radar reflectivity), evaluation metrics and the measurements to enhance readability and to make space for more details, especially with respect to data availability for the radar and SCADA data (publicly available data, protected data or similar). If publicly available, please also state the access point of the data and provide more details about the wind farm.

• We agree with the referee to split the methodology section for better readability. In the revised manuscript we split the methodology into 3 sub sections, as: Common model setup, individual run setups, and performance metrics and observations. We also clarify that SCADA and radar data are currently data-protected under a non-disclosure agreement. Unfortunately, this agreement also disallows revealing specifics of the wind farm considered.

**The following lines are added to highlight data-protection in the methodology section:**

Line 128: The selection of the events in this study is motivated by the occurrence of fast changes in wind direction accompanied by severe yaw misalignment leading to significant power loss as observed by a Belgian offshore wind farm in the North Sea. The methodology utilized to identify these events modifies the approach defined by Hannesdóttir and Kelly (2019) to include yaw misalignment. The

wavelet analysis considers a minimum threshold to identify anomalous changes in wind direction accompanied by severe yaw misalignment experienced by several wind turbines. Severe yaw misalignment potentially has adverse effects on the operational lifetime and fatigue loading of a wind turbine (Wan et al., 2015; Bakhshi and Sandborn, 2016; Laino and Hansen, 1998; Damiani et al., 2018), highlighting its importance and relevance in this study. The SCADA analysis for the identification of these events includes confidential error codes and data that are protected under a non-disclosure agreement, therefore no further details can be provided herein.

**Highlighting non-disclosure on radar data:**

Line 137: Three case studies are considered in this sensitivity analysis, namely, Storm Ciara on 10 February 2020, a low-pressure system on 24 December 2020, and a trough passage on 27 June 2020. The radar data presented therein is not publicly available, but was retrieved through a bilateral agreement with the Royal Meteorological Institute of Belgium (RMI-B). A brief synopsis of these events is presented in the following sub-sections.

**Remarks addressing specific lines**

**Comment 1:** Line 30: The reference (Skamarock et al. 2008) points towards Version 3 of the WRF model, but in your methodology section, you mention that you are using Version 4.2. Is there a particular reason why the Version 3 reference is used here and not Version 4?

• This is indeed incorrect. The reference for WRF has been updated to point to the right reference.

The following lines have been updated in the introduction of the revised manuscript: Line 34: This study utilizes the public domain Weather Research and Forecasting - Advanced Research WRF (WRF-ARW) model developed by the National Center for Atmospheric Research (Skamarock et al., 2019; Powers et al., 2017).

**Comment 2:** *Reference(s) for the statements/quoted numbers of wind gust, travel path, effects etc. are missing. Please consider adding.*

• This is correct, we update the revised manuscript to the correct reference, RMI-B.

**The following lines have been updated in the description of Storm Ciara in the revised manuscript:**

Line 142: Storm Ciara is one of the first extratropical cyclones to hit the European continent in the year 2020, occurring on 10 February 2020 over the Belgian North Sea. Storm Ciara originated in the Atlantic Ocean, moving from the North American continent (starting 3 February 2020) to the European continent (16 February 2020). Storm Ciara swept across the majority of western Europe including United Kingdom and Norway, bringing in heavy precipitation and strong winds with a maximum recorded wind gust of 219 km h-1 at Cap Corse, Corsica, France (footnote 1). Over Belgium, the RMI-B (footnote 2) reported wind gusts of up to 115 km h-1 in Ostend, located at the Belgian coast, with heavy precipitation accompanied by strong winds and thunderstorms.

Adding the footnote 1: https://www.meteo-paris.com/actualites/retro-meteo-2020-les-evenements- climatiques-marquants-en-france, website consulted on 21 April 2022. Adding the footnote 2: https://www.meteo.be/nl/info/nieuwsoverzicht/storm-ciara, website consulted on 21 April 2022. **Comment 3:** "Subsequently, the model is run ...". Please consider reformulation since the current formulation could be misinterpreted as two separate independent simulations (one 24h long simulation and one 21h long simulation). I assume the WRF simulation has been run in one continuous block?

• The simulations have been runs as a continuous block.

We highlight this point in the common WRF setup section:

Line 185: The simulations have been performed as a continuous run including spin-up and evaluation periods.

**Comments 4:** *Line 153/154: "The land surface interactions are kept constant". Please reformulate since it is the parameterization scheme that is kept constant, not the interactions themselves.*

• We thank the referee for pointing this out, the phrasing of these sentences has been changed to reflect these comments.

**We highlight this point in the common WRF setup section:**

Line 189: The Rapid Radiative Transfer Model (RRTMG) (lacono et al., 2008) for longwave and shortwave radiation physics is used by all simulations. Similarly, the land–surface interactions are defined by the unified Noah land surface model (Tewari et al., 2004). The PBL, cumulus, and microphysics schemes are varied amongst the mentioned options as described in Table 1 (in the revised manuscript).

**Comment 5:** *Line 213: Make sure that you are talking about the correct boundary conditions (temporal resolution of the LATERAL boundary conditions not the INITIAL boundary conditions). Please correct.*

• We thank the referee for pointing this out. Throughout the revised manuscript we have adopted the term "update interval of lateral boundary conditions" to clearly outline what is meant, and remove any possible connotations with temporal resolution of the ERA5 product itself (see also first comment to Language corrections below). Since the number of occurrences in the manuscript are high, this is not presented in detail here.

**Technical corrections**

Language corrections:

We thank the referee for their critical review on the manuscript's written English. The typos, comma style, abbreviation definitions and technical detail have been corrected in the revised manuscript. In addition to the specific points raised by the referee, we have thoroughly revised the use of English throughout the manuscript with the help of a native English speaker.

We have amended all the textual changes suggested by the referee. For brevity, we only mention the ones that require further explanation.

• Line 5: while the resolution of the reanalysis products is important, it is the temporal resolution of the lateral boundary condition updates that is investigated here (which is not necessarily identical), so I would suggest a formulation like "update interval of lateral boundary conditions" instead of "temporal resolution of re-analysis data".

Indeed, it is the update interval of the lateral boundary conditions that are varied and not the temporal resolution of initial ERA5 re-analysis itself. We correct for this throughout the revised manuscript.

• Line 34: I am not sure what you mean by "expanse of physics parameterizations". Do you mean variety? I would suggest reformulation.

Yes, we mean the different available options. This sentence is modified as: Line 38: Therein, an array of physics parameterizations and model parameters are available to adequately represent a local weather system.

• Line 62: "[...] explores; redistribution [...]". Sentence structure is unclear, please consider reformulation.

**The sentence is revised as follows:**

Line 82: The GF cumulus parameterization (Grell and Freitas, 2014) is an adjustment type parameterization that redistributes compensating subsidence derived from GD-3D to neighboring grid cells using a Gaussian distribution function and adapts the scale-aware cloud representations from Arakawa et al. (2011).

• Line 333: Unclear what "wind-farm power excursions" means. Please reformulate.

We replaced this term with "variability in power production".

**Tables**

**Comments (1-3):** Table 1: Number of nested domain is misleading. Following WRF naming conventions (see e.g. http://dx.doi.org/10.5065/1dfh-6p97), it would be one parent domain (d01) and 3 nested domains (d02-d04). Furthermore, it would be good to mention if 1-way or 2-way nesting has been used.

Table 1: Time-step information incomplete. It is not clear which domain uses the 20s time step. I would suggest to either mention the time-steps for the four domains explicitly or state the domain which uses the 20s time step and provide the time step ratio.

Table 1: Consider replacement of "update frequency" with "update interval" to be consistent with the values given in the second column (1h and 3h are intervals, not frequencies)

- We thank the referee for pointing out discrepancies and indicating opportunities to improve our tables. The table 1 has been updated as shown below to comply with these suggestions.
- Further, we mention 1-way nesting at:

Line 176: The common model parameters considered for all WRF simulations are summarized in Table 1 (presented below). The baseline horizontal grid spacing of the parent domain d01 is 27 km, while the 1-way nested domains are sequentially refined by a factor of 3 ...

Line 180: A time step of 20 s is considered for parent domain d01, while the time step of nested domains is sequentially refined by a factor of 3.

| Numerical setup                        |                                                                     |
|----------------------------------------|---------------------------------------------------------------------|
| Nested domains (1-way nesting)         | 4                                                                   |
| Horizontal grid spacing                | 27 km (d01) × 9 km (d02) × 3 km (d03) × 1 km (d04)                  |
| Terrain following vertical levels      | 57                                                                  |
| Model top pressure                     | 1000 Pa                                                             |
| Time-steps for domain configuration    | 20 s (d01), 6.67 s (d02), 2.22 s (d03), 0.74 s (d04)                |
| Spin-up period                         | 24 h                                                                |
| Lateral & boundary conditions          | ERA5 reanalysis                                                     |
| Evaluation time, additional to spin up | 21 h (Storm Ciara) and 6 h (Low-pressure system and trough passage) |
| Domain size                            | type 1 / type 2 (Fig. 4)                                     |
| Boundary update interval               | 1h / 3h                                                             |
| Physics parametrizations               |                                                                     |
| Radiation                              | RRTMG radiative                                                     |
| Land surface                           | unified Noah land-surface                                           |
| PBL                                    | MYNN / Shin-Hong / Zhang                                            |
| Microphysics                           | WSM5 / Thompson / Morrison                                          |
| Cumulus                                | KF / GD-3D / msKF / GF                                |

**Comment 4:** Units for MAEs, NEDs and Kantorovich distances are missing, please add. I would also suggest to change the E notation of the Kantorovich distance (rather uncommon in scientific literature) to decimal notation. The differences in scales of magnitude are not large enough to justify such notation to the expense of readability.

• All tables have been updated to indicate metric units. As discussed in the overarching comments, we focus on MAE and omit the Kantorovich distance in the revised manuscript.

**Figures**

In general, the font sizes in some of the figures are too small (especially Figure 1, 2 and the time series plots). Please adjust to increase readability.

Figure 1: Mismatch in time statement in caption (4:00) and reference in main text (4:40,1. 137). Please double-check. A description of the meaning of the star-symbol and the red lines is missing in the figure caption (and other figures captions as well). Please also consider changing the coastline color, which is too similar to colors used in the color map of the variable you are plotting.

*Figure 1,6,8,10,11: Since the focus region is rather small, please add latitude and longitude information to make it easier to locate features.*

Figure 3: insert "are" before "shown"

• We thank the referee for their suggestions on how to improve the quality of our figures. We have incorporated all these in the revised manuscript. Presented below are some example plots (Figures 19 in the revised manuscript) to illustrate the changes implemented.

---

## Author Comment (AC3)

**Author comments in reply to anonymous referee #1**

We thank the anonymous referee #1 for their positive feedback on our revised manuscript. We very much appreciate the time invested in reviewing our work. We thank the referee for their critical and thorough review. We have addressed each of the referee comments as detailed point by point below, which we believe further improves the quality of the manuscript. For brevity, only the major comments on written English are shown here.

We hope our revised manuscript can be accepted for publication.

Adithya Vemuri, on behalf of all co-authors

**Overall comments**

*I think that there was little consistency across these three case studies for which parameterizations tended to perform better or worse is itself still interesting, and points to the need for more robust statistics with (much) larger sample sizes to make more general conclusions with confidence about a "best configuration" for accurately modeling the wind speed and wind direction at offshore wind farms in that region, or whether the "best configuration" will change based on the weather regime. I would like to see the authors state that caveat more clearly, especially in the Conclusion.*

We thank the referees this feedback, we have added the following lines to emphasize the caveats of the study.

Line 437: The results indicate little consistency across the three EWEs for different parameterizations.

Line 443: Overall, the inconsistency across different EWEs found in the current work suggests that a general best model configuration for the Belgian North Sea does not exist, and that best practices are highly dependent on the weather regime under consideration. However, it is important to note that this conclusion is based on a limited sample of EWEs over a single observation point at the offshore wind farm. To further justify and generalize this conclusion, a much larger sample considering more weather events, more observations, and more model configurations is warranted.

**Minor comments**

**Comment 1 & 2:** *1. Uncapitalize terms that are abbreviated if they are common nouns (e.g., extreme weather events, numerical weather prediction, planetary boundary layer, lateral boundary conditions, mean absolute error, etc.), but retain capitalization only for proper nouns (e.g., normalized Euclidean distance).*

*2. When defining an abbreviation, the term that is abbreviated should be singular. For instance, EWE should stand for extreme weather event. If there are multiple extreme weather events, then use EWEs. The same principle applies to LBC: LBC should mean lateral boundary condition, while LBCs should mean lateral boundary conditions. Once this change is made, ensure that all subsequent mentions of the abbreviation correctly refer to the singular (EWE, LBC) or plural (EWEs, LBCs) usage.*

We thank the referee for pointing this out. Common nouns in the revised manuscript are no longer capitalized. All abbreviations in the manuscript are changed to their singular forms, e.g., extreme weather events (EWEs).

**Comment 3**: *3. Generally speaking, single-digit numbers should be spelled out. For instance, "3 PBL parameterizations" should be "three PBL parameterizations" (line 59), "5 classes" should be "five classes" (line 94), and "1-way nested" should be "one-way nested" (line 177). There are exceptions to this general rule, of course (especially when pertaining to things like measured values, time, etc.), but there are quite a few more cases like the three I mentioned that should be corrected.*

We thank the referee for this comment. We have changed the single-digit numbers to their spelled-out forms throughout the manuscript. This is not further shown here.

**Comment 4 to 10:** *4. Lines 42, 221: Delete the "see" before "e.g.". The "see" is already implied.*
*5. Lines 48, 49, 53: Add "the" before "PBL".*
*6. Lines 73–74: "by translating it into compensating subsidence, a combination of vertical advection, moisture, and temperature." — This is awkwardly worded and needs to be revised.*
*7. Line 89: Change "vertical" to "the 3D".*
*8. Line 138: Change "therein" to "herein".*
*9. Line 142: Change "Storm Ciara is" to "Storm Ciara was".*
*10. Line 150: Change "British isles" to "British Isles" (capitalize Isles, as the term as a whole is a proper noun).*

We thank the referee for pointing out the mistakes and missing articles, these have been corrected to the suggested. This is not further shown here.

**Comment 11:** *Figs. 1 and 19: These maps are much improved in this revision. That said, I suggest (but do not require) additional revisions to these plots, specifically the contour intervals. Specifically, I think the upper limit of the range should be something more appropriate to the data (even 10 mm/h barely appears on the filled contour maps, so why is 100 mm/h the upper limit?). I would suggest something like [0.01, 0.10, 0.50, 1.00, 5.00, 10.00, 20.00, 30.00), or whatever set of contour intervals would be helpful to capture the meaningful variations in the data better. This is not a critical fix, but is a principle that can be applied to all sorts of figures; you do not have to stick with uniformly spaced contour intervals, whether in linear or exponential space, if that is not the most meaningful way to display the field. Also, is the star filled white, and thus blocking any view of precipitation behind it, or is it transparent/not filled? Ensuring the star is transparent/not filled white would be a small but important fix.*

We thank the referee for their positive feedback on our revised figures. Thank you for further suggestions. The colors in the figures were plotted using LogNorm in matplotlib with limits, 1e-2 to 100, we have changed this to, 1e-2 to 50 for better readability. The star in the plots has been made transparent as well. The updated the are shown below:

[Figure]

Caption: Figure 1. Observed precipitation rate in mm h−1 provided by a C-band Doppler radar located in Jabbeke on the Belgian coast. The star in the plots represents the offshore wind farm of interest. For the meteorological events: (a) Storm Ciara on 10 February 2020 at 04:40 UTC. (b) Low-pressure system on 24 December 2020 at 02:00 UTC. (c) Trough passage on 27 June 2020 at 15:30 UTC.

[Figure]

Caption: Figure 19. Contours of WRF precipitation rate in mm h$^{-1}$ for the case of Storm Ciara on 10 February 2020 at 04:40 UTC. The plots are presented for cumulus simulation pair H for domain d04. The star in the plots represents the offshore wind farm of interest. (a) Simulation run 9: 1h SH msKF Morrison, on 10 February 2020 at 04:40 UTC (b) Simulation run 10: 1h SH GD-3D Morrison, on 10 February 2020 at 04:40 UTC (c) Simulation run 11: 1h SH GF Morrison, on 10 February 2020 at 04:40 UTC.

**Comment 12:** *Line 157: Note that the low in Fig 2b is centered over the English Channel and Normandy, not over the North Sea.*

This has been corrected in the revised manuscript.

Line 157: Fig. 2b indicate the presence of a low-pressure system over the English Channel and Normandy.

**Comment 13:** *13. Fig. 3: Because you made the right axis red to match the line color, you should do the same for the left axis and data series (either make both blue or both black). Also, the green dotted lines are tough to see. Maybe shade that strip green instead? Or make the linewidths thicker.*

Thank you for pointing out the mismatch and low visibility of green lines. We have chosen to keep SCADA data as blue to be consistent throughout the manuscript. The replotted Fig.13 is shown below:

[Figure]

**Comment 14:** *14. Lines 178–179: Change "terrain following pressure levels" to "terrain-following model levels".*

The following changes were made:

Line 179: In the vertical direction, 57 terrain-following model levels are considered …

**Comment 15:** *15. Fig. 4: Delete "and WRF Post-processing System (WPS)", as it is unnecessary (it is also the WRF Pre-processing System, anyway). You can either crop the title out of the figure to avoid needing to define WPS here, or you can modify the title in the plotgrids NCL script to something like, "WRF Domain Configuration".*

We have chosen to crop the picture, the updated Fig. 4 and caption as shown below:

[Figure]

Caption: Figure 4. WRF nested domain configuration (one-way nesting) considered common to all simulation runs in this study.

**Comment 16:** *16. Table 1. In the row for horizontal grid spacing, use commas instead of x symbols to separate the values of the different domains, as you already did for the time steps a few lines below.*

The table has been updated to with commas instead of x, not further shown here.

**Comment 17:** *17. Line 194: Singular-plural agreement: It should either be "a combination…is considered" or "combinations…are considered".*

Thank you pointing out the mistake, the sentence has been corrected to:

Line 194: … WRF physics parameterizations and options available, a combination of different simulation pairs as described in Table 2 is considered.

**Comment 18:** *18. Line 195: Change "variations of update interval of LBC, PBL, cumulus, and microphysics schemes" to "variations of the update interval of the LBCs, and the PBL, cumulus, and microphysics schemes".*

Thank you pointing out the missing articles, the sentences have been changed to the referee suggestion. This is not further shown here.

**Comment 19:** *19. Lines 212–213: "NED is defined as the resultant of…" — Resultant has a reserved meaning in mathematics, and the right-hand-side of the definition of NED is not a resultant. It is the square root of the sum of squares—maybe there is a fancy mathematical term for that?*

We have refrained from using the term 'resultant' in the revised manuscript, and simply define NED through the equation as

Line 211: To recover a single performance metric, a so-called normalized Euclidean distance (NED) is defined by $NED = \sqrt{MAE_{WDn} + MAE_{WS}}$ , with $MAE_{WDn}$ the normalized MAE of wind direction, and $MAE_{WSn}$ the normalized MAE of horizontal wind speed.

**Comment 20:** *20. Sec. 4.3: It would also be interesting to note that Run 2, while the best member for Case 2, was the worst member for Case 3. Different weather situations lead to different skill for various schemes. And here MYNN PBL was not that bad (or at least not as consistently poor as in other cases).*

Thank you for the suggestions, we have included this in the revised manuscript. The following lines were added:

Line 264: Interestingly, the best-case setup for the low-pressure system case performed the worst for the trough passage case.

**Comment 21:** *21. Line 272: Change "Errorbars" to "Error bars".*

Thank you pointing out the mistake, we have corrected this in the revised manuscript. This is not further shown here.

**Comment 22:** *22. Line 276 and elsewhere: Change "observations" to "results" (or similar). You frequently use "observations" or "is observed" in this manuscript to describe what you noticed in a figure or table, but that is confusing when "observations" elsewhere refers to measurements used as truth for validation. You should generally find other words than "observations" or "is observed" when describing the results or conclusions you saw or determined from figures, tables, etc.*

Thank you pointing this out, we have updated the revised manuscript in places where we could change the words observation/observed to more appropriate ones. For brevity, this is not further shown here.

**Comment 23:** *23. Line 360: Change "schemes" to "scheme".*

Thank you pointing out the mistake, we have corrected this in the revised manuscript. This is not further shown here.

**Comment 24:** *24. Line 367: Change "highlighting" to "highlight". I will also add that the other thing that this finding highlights is the case-to-case variability in skill.*

Thank you for pointing out the mistake and the note, we have included this in the revised manuscript. The following lines were added:

Line 369: This appears to highlight case-to-case variability in skill and the importance of the combination of cumulus and microphysics schemes.

**Comment 25 to 29:** *25. Line 372: Change "better in comparison" to "better by comparison".*
*26. Line 380: Change "3 unique case studies" to "three EWE case studies".*
*27. Line 396: Hyphenate "lower-order".*
*28. Line 405: Add a comma after "(Newman et al., 2022)".*
*29. Line 406: Change "latter raw data" to "raw radar observation data".*

We thank the referee for pointing out the mistakes and missing articles, these have been corrected to the suggested. This is not further shown here.

**Comment 30:** *30. Line 420: "In addition, the simulated precipitation is qualitatively compared to radar data from RMI-B." — Precious little qualitative analysis in precipitation rate was performed here. You simply included Figs. 1 and 19, and stated that precipitation in WRF varied by physics configuration, but*

*you did not describe any qualitative analysis. Either some text should be added in the Discussion (or elsewhere) that better describes your qualitative analysis, or you should remove this part.*

We have rephrased this sentence in the revised manuscript to:

Line 423: Qualitatively, precipitation results are found to be highly sensitive to model setup and type of EWE. No clear tendency towards better accuracy with increased complexity of parameterizations is found.

**Comment 31 to 32:** *31. Line 421: Change "impact of update interval of LBC" to "impact of the update interval of the LBCs". Also change "used for PBL" to "used for the PBL".*
*32. Lines 433–434: Change "combine scale-aware" to "combine the scale-aware". Change "with scale-aware" to "with the scale-aware". Change "and 5-class single moment" to "and the five-class single-moment".*

We thank the referee for pointing out the mistakes and missing articles, these have been corrected to the suggested. This is not further shown here.

---

## Author Comment (AC4)

**Author comments in reply to anonymous referee #2**

We thank anonymous referee #2 for their positive feedback on our revised manuscript and their useful comments on the same. We very much appreciate the time invested on a thorough review of our work. We have addressed each of the referee comments as detailed point by point below, which we believe address all of the reviewer comments. For brevity, comments on technical English are not shown here but will be indicated in the mark-up.

We hope our revised manuscript can be accepted for publication.

(Throughout this document, specific modifications to the revised manuscript are shown in blue)

Adithya Vemuri, on behalf of all co-authors

**Specific comments**

*Language corrections:*

**Comment 2:** *l. 267: "underpredict wind direction" → I am not sure what "underpredict" means in the context of the circular wind direction variable. I would suggest using something like cardinal directions (e.g. southwards shift or something like that) or terms like counter-clockwise shift to indicate the direction of the bias in wind direction.*

We have rephrased the statement to better point out the intended message.

Line 268: Qualitatively, simulation runs 1 through 11 underpredict the fast changes in wind direction for the evaluation period, whereas wind speeds are underpredicted by all runs.

*Tables:*

**Comment 6:** *Table 1: I think "physics parameters" is a bit misleading, since it is not single parameters but whole schemes/parameterisations that are changed as well. I would suggest something like "model settings and physics parameterizations".*

The caption for Table 1 has been updated to:
WRF model setup and common parameters for all simulation runs. The varied model settings and physics parameterizations are highlighted in italics. Scale-aware physics parameterizations are underlined.

*Figures:*

**Comment 7:** *Figure 5, 6, 7: The gray lines of the individual ensemble members are very difficult to see and can be easily confused with the grid lines of the plot. I would suggest to make them a bit thicker or in a color of higher contrast.*

Thank you for the suggestion, we agree that the individual gray lines are difficult to see. However, making the individual ensemble members thicker resulted in a very disorderly figure. As a compromise, we have updated the plots to include the min-max envelope of the ensemble members, as indicated by the examples shown below.

**Comment 8:** *Figure 6, 7, 13: It is very unusual within the wind energy community to represent wind direction in negative degrees (also since Figure 5 seems to use the 0 to 360 deg convention). Consider adapting to the 0 to 360 deg wind direction convention in the plots for consistency and readability.*

Thank you for the suggestion, we have updated the plots to in a better manner, as indicated by the examples shown below.

**Comment 9:** *Figure 5,6,7,13: While it is indirectly implied what the x-axis values mean, it would greatly improve readability and minimize confusion if a x label similar to Fig. 3 could be added.*

Thank you for the suggestion, we have updated the plots to in a better manner, as indicated by the examples shown below.

[Figure]

**Comment 10:** *Figure 7: Caption text ("best-case setup simulation run 2") and figure label do not match (Best-case run 12). Please correct.*

Thank you for pointing out the mistake, we have corrected the figure caption (shown here, an example for trough passage case):
Figure 7. Timeseries plots wind direction and wind speed plotted along with the ensemble average and best-case setup simulation run 12 for the case of trough passage. The minimum and maximum envelope of ensemble members is highlighted in light red. (a) Wind direction. (b) Wind speed.

**Comment 11:** *Figure 8,9,10,11,12: While it was mentioned in the float text, I think it would be good to add also in the figure caption how the error bars are defined.*

Thank you for the suggestion, we have updated the captions of these figures to include the definition of error bars. For brevity, caption for Figure 8:
Figure 8. Performance evaluation for simulation pair A considering change in update interval of LBCs, as described in Table 2. Error bars indicate one standard error of the sample mean. (a) MAE comparison for wind direction. (b) MAE comparison for wind speed.

---

## Referee Report (RR1)

**Referee comment to revised version of**

**Title: Sensitivity analysis of mesoscale simulations to physics parameterizations over the Belgian North Sea using WRF-ARW**
Author(s): Adithya Vemuri et al.

**General comments:**

Firstly, I would like to thank the authors for the very detailed, structured and polite replies to the previous comments. All my previously stated questions and remarks have been answered adequately and the revised manuscript version has improved significantly. The additional work is highly appreciated. Since there have been larger changes in the paper structure and figures, I found some additional minor comments to the revised manuscript (outlined below), especially for the time series figures. Generally however, I would suggest acceptance with minor revisions.

**Specific comments:**

Language corrections:

l. 124: Section 2 → Sect. 2 [for consistency with the following sentences], similar also for l. 232

l. 267: "underpredict wind direction" → I am not sure what "underpredict" means in the context of the circular wind direction variable. I would suggest using something like cardinal directions (e.g. southwards shift or something like that) or terms like counter-clockwise shift to indicate the direction of the bias in wind direction.

l. 367: "to highlighting" → "to highlight"

l. 383: "." too much before "("

l. 435: "," too much before scale-aware SH PBL

Tables:

Table 1: I think "physics parameters" is a bit misleading, since it is not single parameters but whole schemes/parameterisations that are changed as well. I would suggest something like "model settings and physics parameterizations".

Figures:

Figure 5, 6, 7: The gray lines of the individual ensemble members are very difficult to see and can be easily confused with the grid lines of the plot. I would suggest to make them a bit thicker or in a color of higher contrast.

Figure 6, 7, 13: It is very unusual within the wind energy community to represent wind direction in negative degrees (also since Figure 5 seems to use the 0 to 360 deg convention). Consider adapting to the 0 to 360 deg wind direction convention in the plots for consistency and readability.

Figure 5,6,7,13: While it is indirectly implied what the x-axis values mean, it would greatly improve readability and minimize confusion if a x label similar to Fig. 3 could be added.

Figure 7: Caption text ("best-case setup simulation run 2") and figure label do not match (Best-case run 12). Please correct.

Figure 8,9,10,11,12: While it was mentioned in the float text, I think it would be good to add also in the figure caption how the error bars are defined.

---

## Author Response (AR3)

**Author comments in reply to associate editor**

We thank dr. Andrea Hahmann for her useful comments and time invested in reviewing our manuscript. We very much appreciate her support in this review process. We have addressed each of the comments as detailed point by point below, which we believe further improves the quality of our manuscript. We hope our revised manuscript can be accepted for publication.

Adithya Vemuri, on behalf of all co-authors

**Editorial comments**

**Comment 1:** *Tables 3,4 and 5. It is not possible to use tables with colored cells. Please see the note from the file validation. You will need to find an alternative approach.*

We have changed the format of all tables to follow WES guidelines. As an example, Table 3 is presented below. The colored cells were aimed at emphasizing the case-to-case variability in skill, however this has since been added to the manuscript at line 363 and 431 as suggested by anonymous reviewer 1's 2nd revision comments.

| Simulation run# | ERA LBCs updates | PBL scheme | Cumulus scheme | Microphysics scheme | Wind direction MAE (degrees) | Wind speed MAE ($m\ s^{-1}$) | NED (-) |
|---|---|---|---|---|---|---|---|
| 1 | 3 h | MYNN | msKF | WSM5 | 10.46 | 3.88 | 2.08 |
| 2 | 3 h | SH | KF | WSM5 | 8.48 | 2.57 | 1.51 |
| 3 | 1 h | MYNN | KF | Thompson | 9.26 | 2.72 | 1.63 |
| 4 | 1 h | MYNN | msKF | WSM5 | 8.61 | 2.54 | 1.51 |
| 5 | 1 h | SH | KF | WSM5 | 7.68 | 2.47 | 1.41 |
| 6 | 1 h | SH | KF | Thompson | 8.37 | 2.51 | 1.48 |
| 7 | 1 h | SH | msKF | WSM5 | 6.59 | 1.78 | 1.11 |
| 8 | 1 h | SH | msKF | Thompson | 6.69 | 1.89 | 1.15 |
| 9 | 1 h | SH | msKF | Morrison | 7.17 | 1.89 | 1.20 |
| 10 | 1 h | SH | GD-3D | Morrison | 5.59 | 2.25 | 1.17 |
| 11 | 1 h | SH | GF | Morrison | 7.17 | 2.67 | 1.43 |
| 12 | 1 h | Zhang | GF | Morrison | 8.69 | 1.84 | 1.34 |
| 13 | | Ensemble | | | 5.88 | 2.04 | 1.12 |

**Comment 2***: I was alarmed to see a time step of 20s in D1. You should have been able to run with a time step of 70-100 s in the 27 km domain. It is a little too late for this manuscript, but using a very small time step causes enhanced diffusion and smooths the model results. This is not adequately described in the WRF documentation, but you should be aware for future simulations.*

Thank you for the note. We wanted to limit the scope of this sensitivity to the variation in physics parameterizations alone.

**Minor editorial improvements on technical English**

**Comment 1:** *In my opinion, WRF should not be the subject of a sentence, but "the WRF model" is acceptable. I suggest replacing, e.g., L39 "WRF simulations are found…" with "the WRF model simulations*

*are found…”. L48: “WRF physics parameterizations…” with “Physics parameterizations in the WRF model…” etc.*

We thank the editor for pointing this out. We have corrected the aforementioned statements in the revised manuscript. Not further shown here.

**Comment 2**: *L37: “The microphysical parameterizations….”*

We thank the editor for pointing this out. We have corrected the statement in the revised manuscript. Not further shown here.

**Comment 3:** *When there are multiple citations, they should be ordered by year of publication (unless you have a reason to do it which is obvious): L103: (Hahmann et al., 2015; Giannakopoulou and Nhili, 2014; Carvalho et al., 2012) and in other places in the manuscript.*

The citation order has been changed in the following places:

Line 35: … National Center for Atmospheric Research (Powers et al., 2017; Skamarock et al., 2019).
Line 102: … the local weather systems (Carvalho et al., 2012; Giannakopoulou and Nhili, 2014; Hahmann et al., 2015).
Line 114: … by García-Díez et al. (2013), Mooney et al. (2013), and Stergiou et al. (2017) …
Line 132: … fatigue loading of a wind turbine (Laino and Hansen, 1998; Wan et al., 2015; Bakhshi and Sandborn, 2016; Damiani et al., 2018), …

**Comment 4:** *“In order to” -> To, e.g., L194. There are other “wordy” expressions throughout the manuscript.*

The sentence has been rephrased to:

Line 191: To categorize and distinguish the key …

**Comment 5:** *L196: “The sensitivity to hourly and 3-hourly update intervals of LBCs is assessed”. The subject of the sentence is singular.*

We thank the editor for pointing this mistake. We have corrected the statement in the revised manuscript. Not further shown here.

**Comment 6:** *L320: “of the timeseries for the low-pressure”, BTW, "timeseries" is not an English word.*

We thank the editor for pointing this mistake. We have corrected this for all occurrences of “timeseries” in the revised manuscript. Not further shown here.

**Comment 7:** *L322: “Storm Ciara (not shown).”*

We thank the editor for pointing this out. We have corrected the statement in the revised manuscript. Not further shown here.